# Dormancy-specific imprinting underlies maternal inheritance of seed dormancy in Arabidopsis thaliana

Urszula Piskurewicz[1,2†], Mayumi Iwasaki[1,2†], Daichi Susaki[3], Christian Megies[1,2], Tetsu Kinoshita[3], Luis Lopez-Molina[1,2*]

[1]Department of Plant Biology, University of Geneva, Geneva, Switzerland; [2]Institute for Genetics and Genomics in Geneva (iGE3), University of Geneva, Geneva, Switzerland; [3]Kihara Institute for Biological Research, Yokohama City University, Yokohama, Japan

**Abstract** Mature seed dormancy is a vital plant trait that prevents germination out of season. In *Arabidopsis*, the trait can be maternally regulated but the underlying mechanisms sustaining this regulation, its general occurrence and its biological significance among accessions are poorly understood. Upon seed imbibition, the endosperm is essential to repress the germination of dormant seeds. Investigation of genomic imprinting in the mature seed endosperm led us to identify a novel set of imprinted genes that are expressed upon seed imbibition. Remarkably, programs of imprinted gene expression are adapted according to the dormancy status of the seed. We provide direct evidence that imprinted genes play a role in regulating germination processes and that preferential maternal allelic expression can implement maternal inheritance of seed dormancy levels.

*For correspondence: Luis. LopezMolina@unige.ch

†These authors contributed equally to this work

Competing interests: The authors declare that no competing interests exist.

## Introduction

Mature seeds are the endpoint of embryogenesis and highly resistant structures. In angiosperms, seed development is initiated after a double-fertilization event, which produces the endosperm and the zygote. Arabidopsis mature seeds consist of a desiccated and highly resistant embryo surrounded by a single cell layer of endosperm and an external layer, the testa, consisting of dead integumentary maternal tissues. The endosperm nourishes the developing embryo as both tissues develop. The endosperm is triploid, bearing two maternal genomes and one paternal genome, whereas the zygote is diploid, bearing one maternal and one paternal genome.

Seed germination is a developmental transition that transforms the embryo into a fragile seedling. Unsurprisingly, this process is tightly controlled (*Nonogaki, 2014*; *Yan et al., 2014*). Primary seed dormancy, hereafter referred to as 'dormancy', is a property of freshly produced seeds whereby seed germination does not occur even under otherwise favorable germination conditions (*Chahtane et al., 2016*). Dormancy is a vital trait that prevents germination out of season while maintaining the embryo in a protected state within the dry seed. As they age, dry seeds lose dormancy, a process known as dry after-ripening, i.e. they acquire the capacity to germinate under favorable germination conditions. The time period of dry after-ripening required before the seed acquires the capacity to germinate can be used to define the dormancy levels stored in seeds (*Chahtane et al., 2016*). Unsurprisingly, the trait of dormancy varies markedly among plant species, including among *Arabidopsis* accessions, with important consequences in plant ecology, phenology and agriculture (*Baskin and Baskin, 1998*; *Finch-Savage and Leubner-Metzger, 2006*; *Schmuths et al., 2006*; *Meng et al., 2008*; *Springthorpe and Penfield, 2015*). Indeed, different

*Arabidopsis* accessions produce seeds with low or high dormancy levels. Low dormancy accessions need only a short dry after-ripening time to acquire the capacity to germinate upon imbibition, unlike highly dormant accessions.

Final seed dormancy levels are strongly influenced by the climatic conditions experienced by the mother plant. In particular, cold temperatures lead to higher final dormancy levels in *Arabidopsis* mature seeds (*Kendall et al., 2011*). Interestingly, this response is maternally controlled, and involves the genes *FLOWERING LOCUS C* (*FLC*) and *FLOWERING LOCUS T* (*FT*) (*Chiang et al., 2009*; *Chen et al., 2014*).

Abscisic acid (ABA) is a growth-repressive hormone that is essential to repress the germination of dormant seeds (*Debeaujon and Koornneef, 2000*; *Ali-Rachedi et al., 2004*). Furthermore, upon imbibition, ABA stimulates the expression of *LATE EMBRYONIC ABUNDANT* (*LEA*) genes, whose products promote osmotolerance, and inhibits embryonic lipid catabolism (*Lopez-Molina and Chua, 2000*; *Lopez-Molina et al., 2002*; *Penfield et al., 2006*; *Dekkers et al., 2015*).

In *Arabidopsis*, the endosperm is essential to repress the germination of dormant seeds. Indeed, removing the testa and endosperm upon dormant seed imbibition triggers growth and greening of the embryo (*Bethke et al., 2007*; *Lee and Lopez-Molina, 2013*). Furthermore, removing the testa while leaving the endosperm layer surrounding the embryo does not trigger embryonic growth (*Bethke et al., 2007*). A 'seed coat bedding assay', monitoring the growth of dissected embryos cultured on a layer of dissected endosperms with the testa still attached, showed that the endosperm of dormant seeds is able to block embryonic growth by continuously synthesizing and releasing ABA towards the embryo (*Lee et al., 2010*). In fully after-ripened seeds, the endosperm ceases to release sufficient ABA upon imbibition, thus allowing germination to take place (*Lee et al., 2010*).

In summary, dormancy is a multi-faceted process in which, upon seed imbibition, (1) embryonic growth is repressed, (2) utilization of at least part of the seed's food reserves is repressed and (3) osmotolerance gene expression programs are stimulated.

Imprinted gene expression, also called genomic imprinting, is the preferential expression of a given parental allele over the other. Such parent-of-origin gene expression is observed in both mammals and flowering plants, which share the habit of nourishing the embryo through a sexually derived tissue (*Pires and Grossniklaus, 2014*). In *Arabidopsis*, genomic imprinting was found and studied in the endosperm during seed development (*Gehring, 2013*). Given the endosperm's nourishing role, it is often regarded as a plant equivalent of the mammalian placenta. The kinship or 'parental conflict' theory is often proposed to account for the evolutionary origin of imprinting (*Haig and Westoby, 1989*). Nevertheless, the evolutionary forces that led to imprinting remain obscure (*Rodrigues and Zilberman, 2015*).

Dormancy levels can be maternally regulated (*Chiang et al., 2009*; *Chen et al., 2014*). However, the general nature and extent of the maternal control of seed dormancy remain poorly characterized. The occurrence and biological role of endospermic imprinting in *Arabidopsis* mature seeds has not been investigated previously. Given the central role played by the mature endosperm in preventing the germination of dormant seeds, we investigated whether genomic imprinting could be implicated in maternal inheritance of primary seed dormancy. Here, we provide direct evidence that the maternal inheritance of seed dormancy does indeed involve dormancy-specific genomic imprinting programs that take place in the mature endosperm.

## Results

### Maternal inheritance of seed dormancy levels

When cultivated under standard laboratory conditions, *Arabidopsis* accessions Cape Verde Islands-0 (Cvi) and C24 (C24) produce seeds with high and low dormancy levels, respectively. Cvi seeds require a longer dry after-ripening time than C24 seeds to acquire the capacity to germinate. To assess whether seed dormancy levels are determined by parental inheritance in these accessions, we measured the dormancy levels of F1 hybrid seeds generated after reciprocally crossing Cvi and C24 plants. F1 hybrid seeds produced by C24 mother plants (with a Cvi pollen donor) are referred to as C24xCvi F1 seeds and those from Cvi mother plants (with a C24 pollen donor) as CvixC24 F1 seeds (*Figure 1—figure supplement 1*).

Following a short after-ripening period (five days), C24, Cvi, and F1 seeds were unable to germinate 72 hr after seed imbibition (*Figure 1A*). As expected, a long dry after-ripening period (6 months) led to full germination of all seed groups (*Figure 1A*). An intermediate after-ripening time (25 days) led to full germination of C24 seeds but not of Cvi seeds, consistent with their different natural seed dormancy levels (*Figure 1A*). Interestingly, C24xCvi F1 seeds almost fully germinated, unlike CvixC24 F1 seeds, indicating that C24xCvi F1 seeds are less dormant than CvixC24 F1 seeds (*Figure 1A*). In all cases where seeds were dormant, removal of the testa and endosperm triggered embryonic growth, consistent with previous results (*Figure 1A*) (*Bethke et al., 2007*; *Lee et al., 2010*).

Similar observations were made with CvixCol and ColxCvi F1 seeds generated after reciprocally crossing Cvi plants with the low dormancy accession Columbia-0 (Col) (*Figure 1C*, *Figure 1—figure supplement 3*).

At least for the particular ecotypes studied here, these observations indicate that hybrid F1 seeds tend to inherit dormancy levels more akin to their maternal genotype, which necessitates the endosperm's germination repressive activity. Maternal inheritance of dormancy levels could be due to several factors including: (1) a purely maternal effect whereby the Cvi maternal plant tissues impose higher dormancy to the seed progeny; (2) an endospermic gene-imprinting effect; (3) a gene expression dosage effect resulting from the endosperm's parental genome imbalance, which favors the maternal genome (2 maternal vs. 1 paternal genome, *Figure 1—figure supplement 1*). In the last case, duplication of maternal dormancy genes would impose a pattern in their expression more akin to the maternal genotype. These three possibilities are not mutually exclusive and could therefore all be valid.

To assess the contribution of endospermic gene dosage effects, we used tetraploid Cvi-0 (Cvi[tet]) and C24 (C24[tet]) plants as pollen donors in crosses with normal diploid C24 and Cvi female accessions, respectively. These crosses produce hybrid C24xCvi[tet] and CvixC24[tet] F1 seeds containing a 2:2 Cvi to C24 genome ratio in endosperm cells (*Figure 1—figure supplement 2*). Unsurprisingly, Cvi[tet] seeds were more dormant than C24[tet] seeds (*Figure 1B*). Furthermore, CvixC24[tet] F1 seeds were more dormant than C24xCvi[tet] F1 seeds (*Figure 1B*). As expected, presence of the endosperm was required to repress seed germination (*Figure 1B*). These data therefore indicate that the endosperm's natural maternal genome imbalance does not readily account for the observed inheritance in F1 seeds of dormancy levels akin to those of their maternal genotype (*Figure 1A*). This prompted us to explore whether F1 seed dormancy is associated with genomic imprinting in the endosperm of F1 seeds.

## The mature testa contains insignificant amounts of mRNA relative to the endosperm

Partial dissection of the endosperm, i.e. the isolation of the endosperm with the testa still attached to it, is a rapid and easy procedure. By contrast, full dissection of the endosperm, i.e. the isolation of the endosperm without the surrounding testa, is a challenging task because it requires peeling the testa, a delicate and lengthy procedure that more often than not wounds the endosperm. The testa originates from ovular integuments that undergo progressive degeneration and collapse during seed development. Electron microscopy studies reveal that the mature testa does not contain obvious traces of organelles, cytoplasm or nuclei (*Yadav et al., 2014*). Accordingly, researchers assume that the testa is not a significant source of maternal RNA contamination and thus that the partially dissected endosperm can be used as a source of RNA for endosperm transcriptomic studies (*Dekkers et al., 2013*; *Lee and Lopez-Molina, 2013*).

We sought to verify that the testa is not a significant source of maternal mRNA contamination. Using WT seed material, total RNA was isolated from 40 partially dissected endosperms, 40 testas and 40 fully dissected endosperms in two independent experiments. To ensure maximal RNA recovery, a nucleic acid carrier was used for final RNA precipitation (*Table 1*, Materials and methods). The concentration levels of RNA extracted from testas were below detection levels, unlike those from partially and fully dissected endosperm (*Table 1*). 200 ng of RNA was used to construct partially and fully dissected endosperm cDNA libraries. Although testa RNA could not be detected in our RNA testa sample, we sought to construct testa cDNA libraries using avolume of RNA testa sample equal to that of the largest one used for the construction of the partially and fully dissected endosperm cDNA libraries (*Table 1*, Materials and methods). Unsurprisingly, the resulting testa libraries

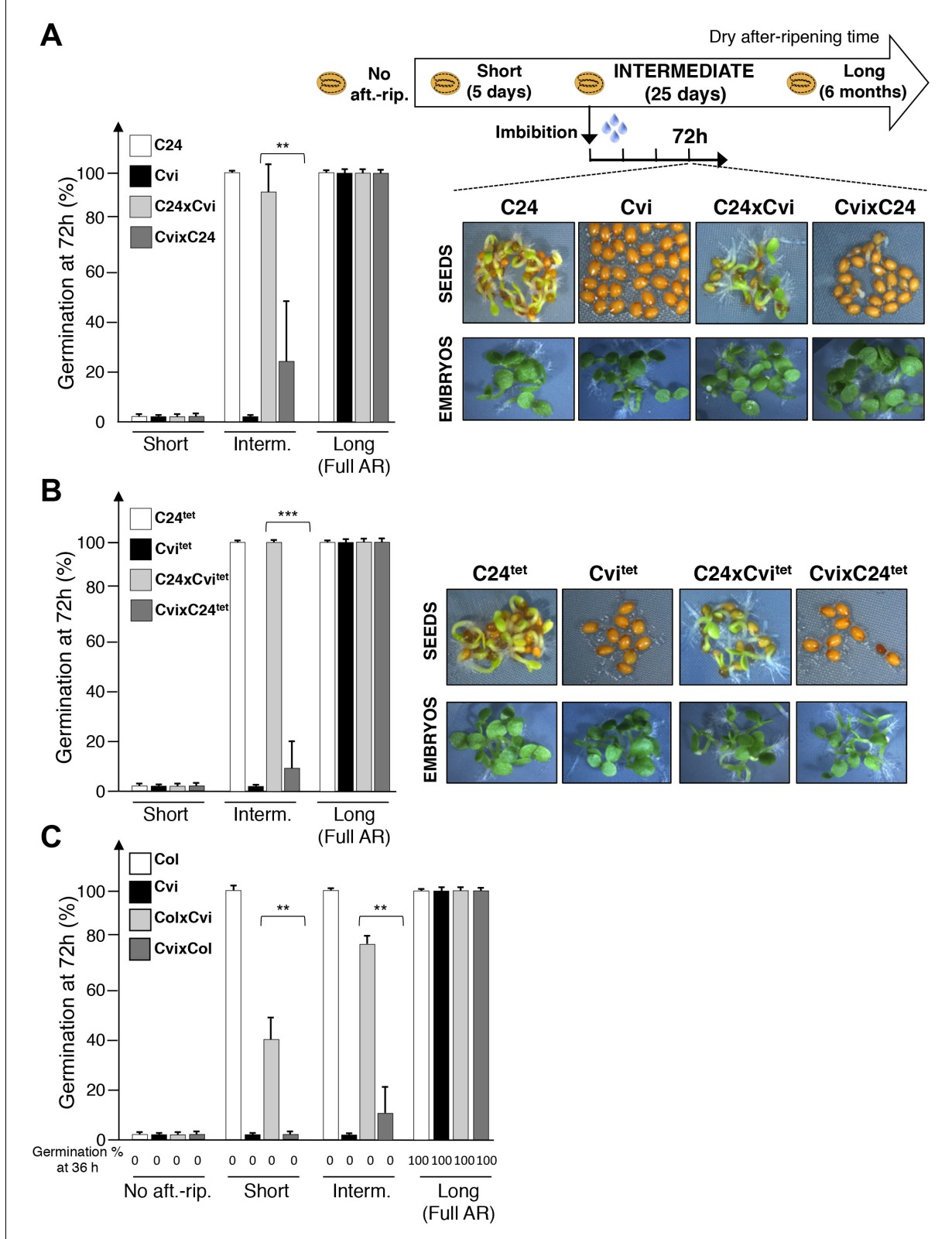

**Figure 1.** Maternal inheritance of seed dormancy. (**A**) Histograms depicting percent germination of C24, Cvi, C24xCvi F1 and CvixC24 F1 seed populations after-ripened for short (five days), intermediate (25 days) or long (six months) time periods. Germination was scored after 72 hr of seed imbibition (4 replicates, n = 150–200, \*\*p<0.01, \*\*\*p<0.001, Student's t test). Pictures show intermediately after-ripened C24, Cvi, C24xCvi F1 and CvixC24 F1 seeds 72 hr after imbibition (upper panel) and the respective embryos (lower panel) following testa and endosperm removal 4 hr after seed

*Figure 1 continued on next page*

*Figure 1 continued*

imbibition. (**B**) Same as (**A**) using C24[tet], Cvi[tet], C24xCvi[tet] F1 and CvixC24[tet] F1 seeds. (**C**) Same as (**A**) with Col, Cvi, ColxCvi F1 and CvixCol F1 seeds that were not after-ripened (No aft.-rip) or after-ripened for short (10 days), intermediate (20 days) and long (six months) time periods.

The following figure supplements are available for figure 1:

**Figure supplement 1.** Procedure for assessing CvixC24 F1 and C24xCvi F1 hybrid seed production, dry after-ripening and seed germination.

**Figure supplement 2.** Procedure for the assessment of CvixC24[tet] F1 and C24xCvi[tet] F1 hybrid seed production, dry after-ripening and seed germination.

**Figure supplement 3.** Maternal inheritance of seed dormancy.

contained concentrations of cDNA that were smaller than 25 fold than those in the partially and fully dissected endosperm (*Table 1*). All cDNA libraries were subject to high-throughput sequencing and sequencing reads were mapped to the *Arabidopsis* genome. The total number of reads obtained with the testa sample was only about 1% of the read numbers obtained for the partially and fully dissected endosperm (*Table 1*). By contrast, the read numbers of the partially and fully dissected endosperm samples were similar (*Table 1*). We compared the gene expression profiles between partially and fully dissected endosperm samples. Only 14 genes among 33,557 annotated genes were differentially expressed between the partially and fully dissected endosperm samples ($p < 0.05$) (*Figure 2—source data 1*) (Material and methods). Among these 14 genes, seven are osmotic stress-induced genes whose expression was higher in the fully dissected endosperm relative to the partially dissected endosperm (*Figure 2—source data 2*) (*Kilian et al., 2007*; *Winter et al., 2007*). The differential expression of these genes could be due to the lengthy testa peeling procedure, prior to endosperm isolation, which likely deprives the endosperm tissue of moisture. Similarly, among the 14 genes, two genes (*AT4G35100* and *AT1G62510*) were reported to be downregulated in response to osmotic stress and had higher expression in the partially dissected endosperm relative to the fully dissected endosperm (*Figure 2—source data 1*) (*Kilian et al., 2007*; *Winter et al., 2007*). Only *AT1G62510*, predicted to encode a seed-storage 2S albumin-like protein, had a substantial number of reads in the testa sample (*Figure 2—figure supplement 1*). This gene is expressed during seed maturation, suggesting that a maternal contamination could also account for its higher expression in the partially dissected endosperm relative to that in the fully dissected endosperm (*Winter et al., 2007*).

Altogether, these results confirm the notion that low amounts of mRNA are present in the testa relative to the endosperm and that the partially dissected endosperm of mature seeds can be used

**Table 1.** RNA concentrations and read number after RNA-seq using RNA isolated from partially dissected endosperm (with testa still attached to the endosperm), fully dissected endosperm, and testa.

| Sample | RNA concentration ng/µl in 50 µl volume | Total amount of RNA (ng) | Volume of RNA sample used for library preparation (µl) | cDNA library (ng/µl) | Volume of library used for sequencing | Total mapped sequencing reads |
|---|---|---|---|---|---|---|
| Partially dissected endosperm_A | 5 | 250 | 40 | 37.4 | 1.16 | 39,097,682 |
| Partially dissected endosperm_B | 6.4 | 320 | 31.25 | 15.2 | 2.5 | 31,930,392 |
| Fully dissected endosperm_A | 7 | 350 | 28.57 | 16.4 | 2.5 | 39,862,912 |
| Fully dissected endosperm_B | 4.4 | 220 | 45.45 | 15.8 | 2.5 | 35,254,459 |
| Testa_A | 0 | 0 | 45.45 | 0.37 | 2.5 | 264,057 |
| Testa_B | 0 | 0 | 45.45 | 0.59 | 2.5 | 595,710 |

as a source of endospermic mRNA without a major concern of maternal mRNA contamination (see also below). Furthermore, the contamination of partially dissected endosperm samples with embryonic tissue is not a major concern (*Lee et al., 2012*).

## Dormancy-specific genomic imprinting in the endosperm

Hereafter, for simplification purposes and unless otherwise specified, we use the term 'endosperm' for 'partially dissected endosperm'.

ColxCvi and CvixCol F1 seeds were harvested the same day and after-ripened for 10 days (dormant seeds) or for two months (non-dormant seeds). Total RNA samples were extracted from the endosperms (n = 200) of dormant and non-dormant CvixCol and ColxCvi F1 seeds 36 hr after seed imbibition (*Figure 2A*). At this time, CvixCol and ColxCvi F1 seeds have not yet germinated, even though ColxCvi F1 seeds are less dormant that CvixCol F1 seeds (*Figure 1C*). Thus, 36 hr after imbibition is a time when the endosperm of ColxCvi and CvixCol F1 seeds ought to express the genetic dormancy programs controlling seed germination that are akin to their maternal genotype. The resulting cDNA libraries were subject to high-throughput sequencing and single nucleotide polymorphisms (SNPs) were used to quantify the expression arising from Col and Cvi gene alleles (Materials and methods). The large majority of genes had a 2:1 ratio of maternal to paternal allele expression (*Figure 2—figure supplement 2*). Hundreds of genes had a high expression bias when present as either Cvi or Col alleles, consistent with the results of previous studies analyzing endosperm and embryo gene expression during early embryogenesis (*Figure 2—figure supplement 2*) (*Nodine and Bartel, 2012*; *Pignatta et al., 2014*).

In the endosperm from dormant CvixCol and ColxCvi F1 seeds, we identified 71 maternally expressed genes (MEGs) and 5 paternally expressed genes (PEGs) (*Figure 2B*, *Figure 2—source data 3*). In the endosperm from non-dormant CvixCol and ColxCvi F1 seeds, we identified 50 MEGs and 8 PEGs (*Figure 2B*, *Figure 2—source data 3*). Only 14 MEGs and 2 PEGs were present in both dormant and non-dormant data sets (*Figure 2B*). To further address potential maternal testa RNA contamination, we counted the read numbers of the identified MEGs in the testa cDNA library discussed above (*Table 1*). Among the 107 MEGs, only two (*AT4G00220* and *AT4G04955*) had at least one read (*Table 2*). However, their total read number was less than 1-2% of that found in the partially or fully dissected endosperm cDNA libraries (*Table 2*). Furthermore, as indicated above, none of the MEGs had expression levels that were significantly different between partially and fully dissected endosperm samples (*Figure 2—source data 2*). These data further strengthen the notion that maternal mRNA contamination from the testa is unlikely.

To validate the occurrence of imprinting, we performed RT-PCR on 17 MEGs and 2 PEGs in independent CvixCol and ColxCvi F1 endosperm material. Amplicons were sequenced by Sanger and MiSeq sequencing (see Materials and methods). The results confirmed the RNA-seq analysis (*Figure 2—figure supplement 3*, *Figure 2—source data 2*). In the particular case of *FWA*, a previous report identified *FWA* genomic sequences as sufficient to confer maternal allele expression during embryogenesis in transgenic experiments (*Kinoshita et al., 2004*). The *pFWA::dFWA-GFP* line was reciprocally crossed with Col and a *pDD65::mtKaede* transgenic line in which GFP is localized in mitochondria (Materials and methods) (*Arimura et al. 2004*). Analysis of the F1 transgenic seed material confirmed the maternal expression of the *pFWA::dFWA-GFP* transgene in mature endosperm (*Figure 2—figure supplement 4*).

We compared the identified endospermic MEGs and PEGs with those reported during early embryogenesis using the same F1 seed material (*Pignatta et al., 2014*). Only 6 MEGs and 3 PEGs found here were previously reported, including the MEG *FWA* (*Figure 2B*) (*Pignatta et al., 2014*).

These results strongly suggest the occurrence of specific programs of endospermic genomic imprinting that take place upon seed imbibition. Furthermore, these imprinted gene expression programs appear to change according to seed dormancy levels. Given that dormancy is maternally inherited and given the small number of identified PEGs, we focused on studying dormancy-specific MEGs. Indeed, among the 57 dormancy-specific MEGs, seven were shown to regulate dormancy or germination, whereas another nine could potentially perform the same role based on current knowledge (*Supplementary file 1* and see below). By contrast, only three genes that directly (two genes) or indirectly (one gene) regulate germination were among the 36 non-dormant-specific MEGs (*Supplementary file 1*).

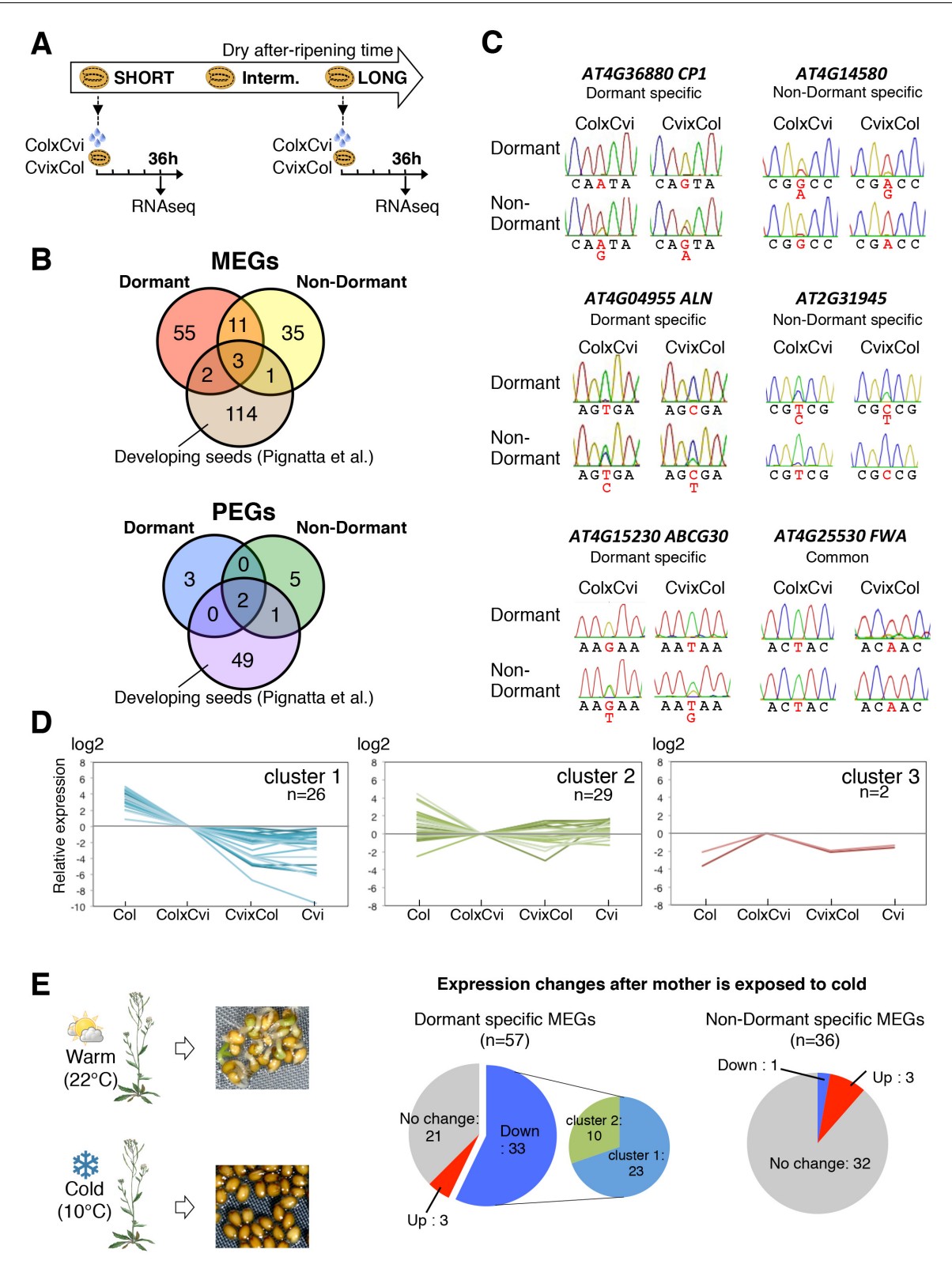

**Figure 2.** Dynamic changes in imprinted gene expression. (A) Seed treatment procedure used to isolate endosperm from seeds. CvixCol and ColxCvi F1 seeds were after-ripened for short (10 days) or long (two months) time periods. RNA that was extracted from endosperm dissected 36 hr after imbibition was processed for RNA-seq and RT-PCR for Sanger sequencing analysis. (B) Overlap among maternally expressed imprinted genes (MEGs) and paternally expressed imprinted genes (PEGs) in the endosperm of dormant and non-dormant seeds and of developing seeds according to

*Figure 2 continued on next page*

*Figure 2 continued*

*Pignatta et al. (2014)*. (**C**) Sanger sequencing chromatograms at SNPs of selected genes. RNA from shortly after-ripened (Dormant) and fully after-ripened (Non-Dormant) CvixCol and ColxCvi endosperms were analyzed. Three Dormant-specific MEGs and two Non-dormant-specific MEGs and one MEG found in all data sets (Dormant, Non-dormant, and Early embryogenesis) are shown. Nucleotides at SNP sites are highlighted in red. (**D**) K-means clusters of dormant-specific MEGs expression levels. RNA samples extracted from endosperms of shortly after-ripened Col, ColxCvi, CvixCol, and Cvi seeds were analyzed. For each gene, expression levels relative to those found in ColxCvi endosperms are shown. (**E**) Gene expression profiles in response to cold treatment of the mother plants during seed development. Left panel: Mother plants exposed to cold temperature produce highly dormant seeds. Right panel: The pie charts group MEGs according to how their expression responds upon seed imbibition after the mother plant was exposed to cold temperatures. 'Down' means that expression was downregulated more than two fold (p < 0.05), 'Up' means that expression was upregulated more than two fold (p < 0.05).

The following source data and figure supplements are available for figure 2:

**Source data 1.** Genes that are differentially expressed between the endosperm+testa and endosperm samples.
**Source data 2.** mRNA expression of MEGs in the testa+endosperm and endosperm samples.
**Source data 3.** Imprinted genes in the endosperm of mature seed.
**Source data 4.** mRNA expression of dormancy-specific MEGs.
**Source data 5.** Expression of MEGs in response to cold treatment during seed development.
**Source data 6.** RNA-seq data for all genes for imprinting in the endosperm of mature seed.
**Source data 7.** Resequencing of amplicons containing SNPs.
**Figure supplement 1.** Histograms depicting the read coverage for the *AT1G62510* and *AT4G35100* genes.
**Figure supplement 2.** Ratio of maternal reads to duplicated paternal reads for endospermic mRNA isolated from CvixCol and ColxCvi F1.
**Figure supplement 3.** Sanger sequencing chromatograms at SNPs of dormant-specific MEGs.
**Figure supplement 4.** Imprinting of *FWA* is observed in the endosperm of mature seed.
**Figure supplement 5.** Comparison of the gene expression profiles of Col and Cvi.
**Figure supplement 6.** Testa removal does not trigger the germination of highly dormant seeds that developed under low temperatures (10°C).

## Dormancy-specific MEG expression can correlate with seed dormancy levels

We evaluated whether dormancy-specific MEG expression correlates with dormancy levels. Clustering analysis of gene expression levels revealed that among the 57 identified dormancy-specific MEGs, about half (26 out of 57) had expression levels that were negatively correlated with seed dormancy levels: Cvi seeds being the most dormant, followed by CvixCol F1, ColxCvi F1 and finally Col seeds, which are the least dormant (cluster 1 in *Figure 2D*, *Figure 2—source data 4*). Strikingly, the majority of MEGs (12 out of 16) that are involved or potentially involved in regulating seed

**Table 2.** Read numbers of the identified MEGs in testa, partially dissected endosperm (endosperm + testa), and fully dissected endosperm samples.

| | Testa | Partially dissected endosperm | Fully dissected endosperm |
|---|---|---|---|
| AT4G00220 | 9 | 603 | 597 |
| AT4G04955 | 3 | 2,154 | 1,859 |

germination belonged to cluster 1 (*Supplementary file 1*). Other gene expression clusters did not show any apparent correlation between mRNA and seed dormancy levels (cluster 2 and 3, *Figure 2D*). Furthermore, among all *Arabidopsis* genes only a minority (about 7%) had expression levels that negatively correlated with seed dormancy levels (*Figure 2—figure supplement 5*). These data therefore indicate that genes belonging to the dormancy-specific MEG group tend to have expression levels that correlate negatively with seed dormancy levels.

Cultivating *Arabidopsis* plants under low temperatures (10°C) increases final seed dormancy levels (*Figure 2E*). Using seed nicking experiments, a previous report indicated that the presence of an intact endosperm is necessary to repress the germination of highly dormant seeds produced under low temperatures (*Kendall et al., 2011*). Indeed, we observed that removal of the endosperm and testa, but not removal of the testa alone, triggered embryonic growth and greening (*Figure 2—figure supplement 6*). This demonstrates that the endosperm is also necessary to repress the germination of highly dormant seeds that developed under low temperatures.

Strikingly, among the 57 dormant-specific MEGs, about half (33) had their expression lowered when seeds developed under cold temperatures (*Figure 2E*, *Figure 2—source data 5*). By contrast, among the 36 MEGs identified in non-dormant seeds, only one had its expression lowered while the expression of the vast majority of genes was unchanged. Furthermore, the majority (23 out of 33) of the genes that were downregulated in response to cold belonged to the cluster 1 of genes whose expression negatively correlates with dormancy levels (*Figure 2E*, *Figure 2—source data 5*).

In summary, at least a third of dormancy-specific MEGs are involved or potentially involved in regulating seed germination. Moreover, about half of dormancy-specific MEGs have their expression correlated negatively with seed dormancy levels. Among the latter, most have their expression downregulated by cold during seed development, which increases mature seed dormancy levels.

We focused our attention on two MEGs, *CYSTEINE PROTEASE1* (*CP1*) and *ALLANTOINASE* (*ALN*). First, we investigated whether these genes play a role in regulating seed germination processes, which has not been investigated previously. Second, we asked whether their maternal allele expression in dormant seeds plays a role in implementing maternal inheritance in seed dormancy.

*CP1* encodes a predicted ortholog of radish DRCP26, which is involved in protein storage decay during the early phase of seed germination (*Tsuji et al., 2013*). We found that *CP1* expression is stringently regulated upon seed imbibition according to seed dormancy levels and cold temperatures during seed development (*Figure 3* and *Figure 2—source data 5*). Furthermore, since food utilization and the imprinting phenomenon were proposed to be closely connected, we reasoned that *CP1* could be a candidate MEG governing seed food store utilization according to maternally inherited dormancy levels.

## Low and preferential maternal *CP1* allele expression is prolonged in dormant seeds

We studied in more detail the dynamics of *CP1* expression according to seed dormancy levels in CvixCol and ColxCvi F1 seeds.

In absence of dry after-ripening, CvixCol and ColxCvi F1 seeds were unable to germinate up to five days after seed imbibition. *CP1* mRNA could not be detected 36 hr after seed imbibition (*Figure 3*).

After a short dry after-ripening period of 10 days, CvixCol and ColxCvi F1 seeds did not germinate two days after imbibition but 60% of ColxCvi F1 seeds germinated five days thereafter while CvixCol F1 seeds did not. *CP1* mRNA expression could be detected 36 hr after imbibition in the endosperm of both CvixCol and ColxCvi F1 seeds, and Sanger sequencing confirmed that it was preferentially maternal (*Figure 3*). Furthermore, *CP1* maternal allele expression was higher in the endosperm of ColxCvi F1 seeds than in that of CvixCol F1 seeds (*Figure 3*).

We next considered an intermediate after-ripening time of 20 days, after which CvixCol F1 seeds were still unable to germinate 2 and 5 days after imbibition, unlike ColxCvi F1 seeds whose populations germinated at about 40% and 100%, respectively (*Figure 3*). In CvixCol F1 seeds, *CP1* expression was higher than that in shortly after-ripened CvixCol F1 seeds (*Figure 3*). However, this expression remained preferentially maternal and lower than that in ColxCvi F1 seeds (*Figure 3*). Furthermore, *CP1* expression was no longer preferentially maternal in ColxCvi F1 seeds.

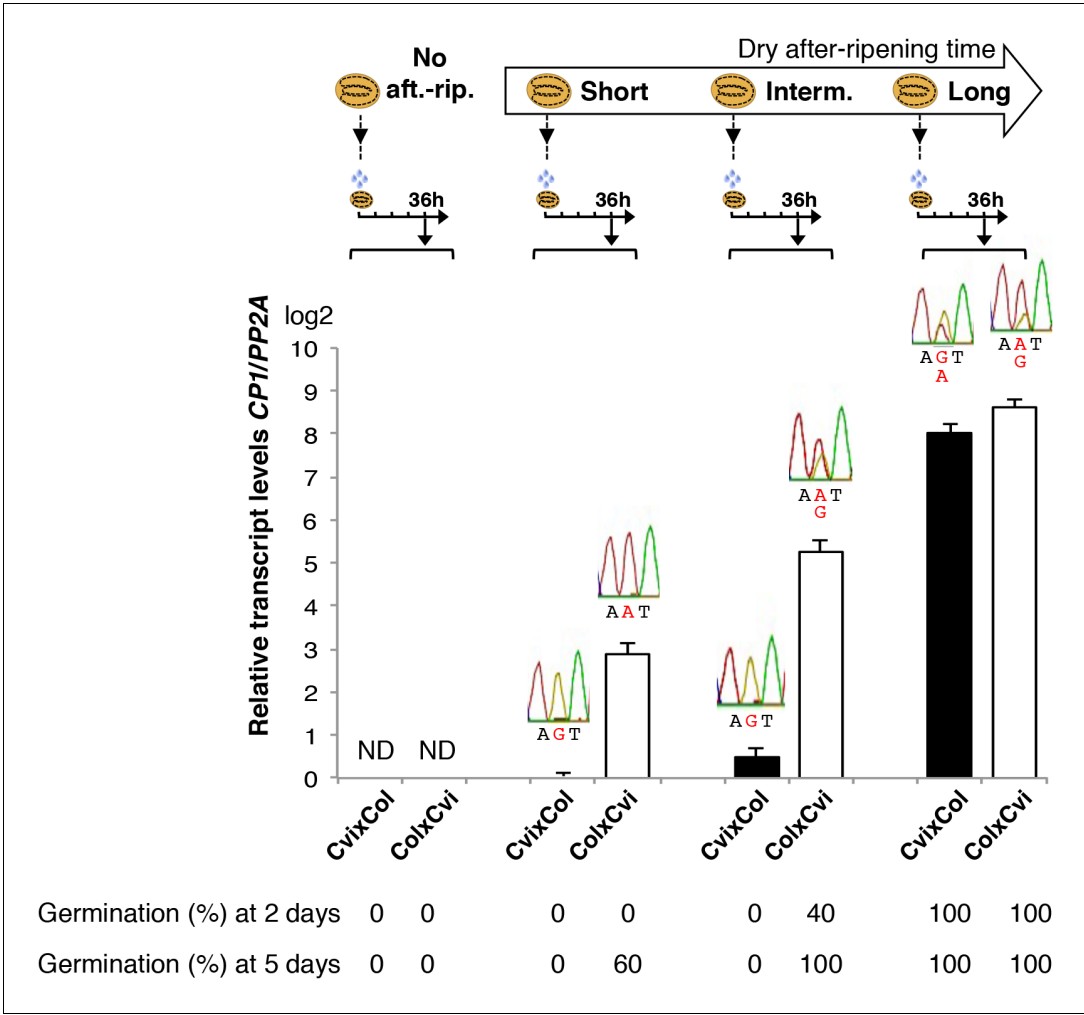

**Figure 3.** Low and preferential maternal *CP1* allele expression is prolonged in dormant seeds but not in non-dormant seeds. CvixCol F1 and ColxCvi F1 seeds were either not after-ripened (No aft.-rip.) or after-ripened for short (10 days), intermediate (20 days), and long (two months) time periods. Percent germination 2 and 5 days after seed imbibition is shown. RNA extracted from endosperm 36 hr after seed imbibition, i.e. prior to germination, was used for qPCR and RT-PCR for Sanger sequencing analysis. *CP1* expression levels relative to those found in the endosperm from CvixCol F1 seeds that had been after-ripened for an indicated time period are shown. *CP1* expression levels were normalized to those of *PP2A*. ND: no detection.

The following figure supplement is available for figure 3:

**Figure supplement 1.** Low and preferential maternal *CP1* alelle expression is prolonged in dormant seeds but not in non-dormant seeds.

Finally, after a long after-ripening period of two months, *CP1* expression was similarly high and biallelic in both CvixCol and ColxCvi F1 seeds (*Figure 3*).

These results confirm that *CP1* preferential maternal allele expression is only found in dormant seeds (*Figure 2C*). Furthermore, the expression levels of maternal *CP1* alleles correlate negatively with dormancy levels. As seeds lose dormancy, *CP1* preferential maternal allele expression is lost and *CP1* expression increases. Similar observations were made with CvixC24 and C24xCvi F1 seeds (*Figure 3—figure supplement 1*).

## Relationship between seed dormancy levels and cruciferin proteins decay upon seed imbibition

*CP1* is predicted to regulate protein storage decay in *Arabidopsis* seeds (*Tsuji et al., 2013*). Cruciferins (CRUs) are highly abundant storage proteins present in endosperm and embryos that can be detected by coomassie dye staining in SDS-PAGE gels (*Barthole et al., 2014*).

We first studied how seed dormancy levels affect protein storage decay using antibodies raised against CRUs and coomassie dye staining (*Figure 4—figure supplement 1*) (*Jolivet et al., 2011*; *Barthole et al., 2014*). In the absence of dry after-ripening, WT CRU levels in the endosperm remained constant over time upon seed imbibition, consistent with the dormant state of the seeds (*Figure 4A* and *Figure 4—figure supplement 2*). By contrast, in fully after-ripened seeds, endospermic CRUs levels rapidly decayed prior to germination (*Figure 4A* and *Figure 4—figure supplement 2*). Following a short after-ripening period that does not fully eliminate seed dormancy, endospermic decay of CRU proteins and germination took place upon imbibition, but both processes were delayed relative to fully after-ripened seeds (*Figure 4A* and *Figure 4—figure supplement 2*). Altogether, these results show that endospermic abundance of CRU proteins upon imbibition reflects seed dormancy levels.

## *CP1* promotes CRU protein decay upon seed imbibition

We next used a genetic approach to ask whether *CP1* is necessary to regulate the decay of CRU proteins using *cp1-1*, *cp1-2* and *cp1-3* mutant seeds bearing a T-DNA insertion in *CP1*'s second exon, second intron and promoter, respectively (Materials and methods). In all of these lines, the decay of CRU proteins was delayed relative to WT in non-dormant seeds (*Figure 4* and *Figure 4—figure supplement 3*). We chose the *cp1-1* mutant line (hereafter referred to as *cp1*) for detailed characterization of CRU protein decay in dormant vs non-dormant seeds.

The abundance of CRUs was similar in WT and *cp1* mutant dry seeds. In absence of WT and *cp1* seed after-ripening, CRUs levels in the endosperm remained constant over time after imbibition (*Figure 4A* and *Figure 4—figure supplement 2*). By contrast, in fully after-ripened seeds, WT endosperm CRU levels rapidly decayed upon imbibition whereas in *cp1* mutants, the decay of CRUs levels was delayed by at least 12 hr (*Figure 4A* and *Figure 4—figure supplement 2*). This delay was not due to slower *cp1* mutant germination as both testa rupture and endosperm rupture events proceeded at the same pace in WT and *cp1* seeds, indicating that *CP1* is not required to regulate germination sensu stricto (*Weitbrecht et al., 2011*) (*Figure 4—figure supplement 4*). Similarly, in shortly after-ripened seeds, CRU protein decay was delayed in *cp1* mutant seeds relative to WT seeds even though *cp1* and WT seed germination proceeded at the same pace (*Figure 4A*, *Figure 4—figure supplement 2* and *Figure 4—figure supplement 4*). During the early stages of germination examined above, no differences in the accumulation of CRUs proteins could be detected between WT and *cp1* embryos (*Figure 4—figure supplement 5*).

Taken together, these data show that *CP1* is necessary to promote the decay of endosperm CRU proteins prior to the germination of shortly and fully after-ripened seeds.

## The decay of CRU proteins is maternally controlled through *CP1*

We next explored whether CRU protein decay is maternally controlled through *CP1* by analyzing CRUs accumulation in hybrid seeds arising from reciprocal crosses between WT and *cp1* plants (*Figure 4—figure supplement 6*).

Only conditions in which *CP1* expression is preferentially maternal upon seed imbibition, as in shortly after-ripened seeds, are expected to be associated with detectable differences in the abundance of CRU proteins in the endosperm of WTx*cp1* and *cp1*xWT F1 seeds upon seed imbibition (*Figure 4—figure supplement 6*). On the other hand, conditions in which *CP1* expression is undetectable, such as the absence of dry after-ripening, or very high and biallelic, as in fully after-ripened seeds, should not be associated with marked differences in CRU abundance between WTx*cp1* and *cp1*xWT F1 endosperm upon seed imbibition (*Figure 4—figure supplement 6*). Consistent with this prediction, in the absence of dry after-ripening or in fully after-ripened seeds, CRU protein levels were similar between WTx*cp1* and *cp1*xWT F1 endosperm 36 hr after seed imbibition (*Figure 4B* and *Figure 4—figure supplement 2B*). By contrast, in shortly after-ripened WTx*cp1* and *cp1*xWT F1 seeds, CRU protein levels were higher in the *cp1*xWT F1 endosperm than in the WTx*cp1* F1

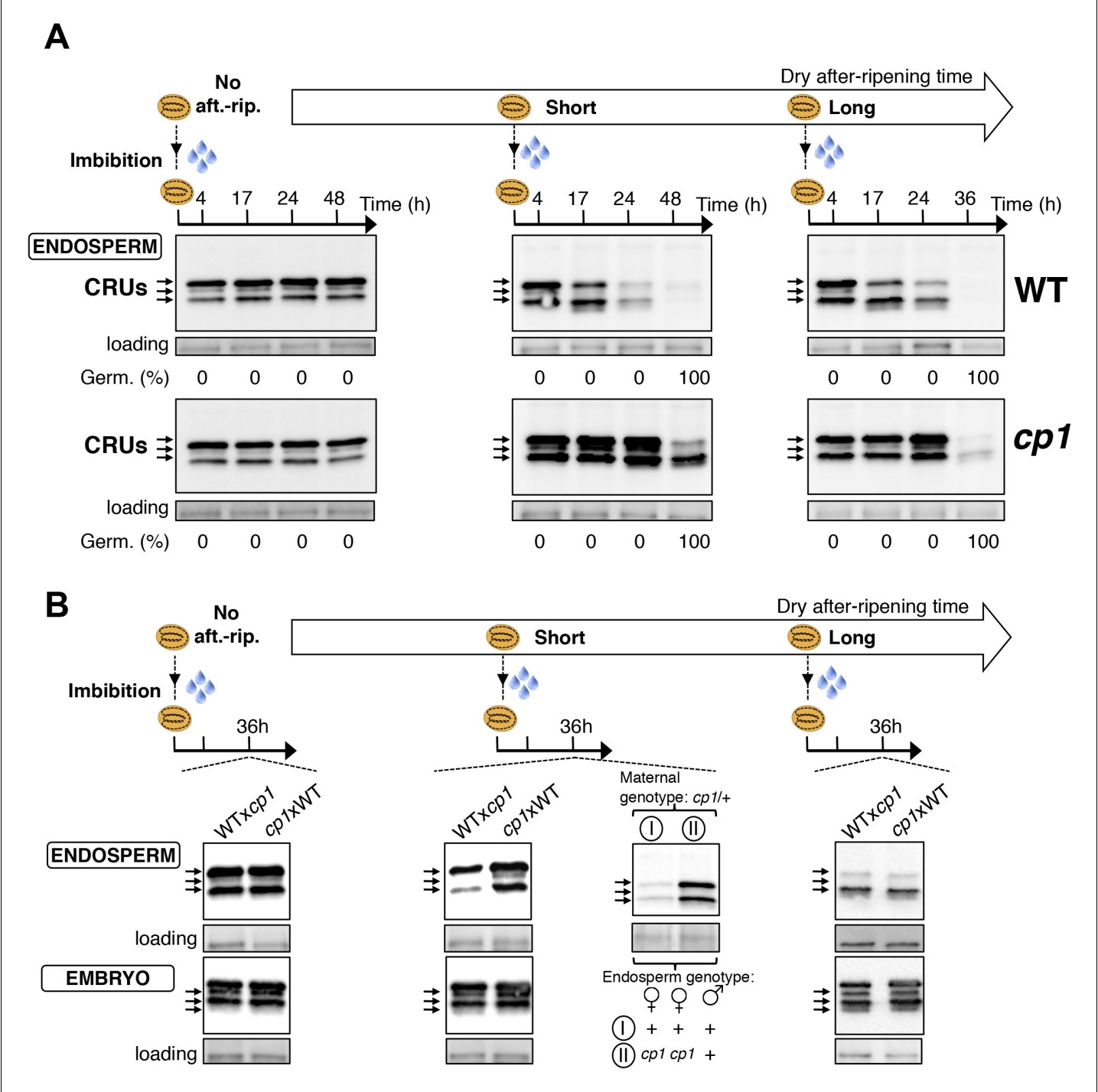

**Figure 4.** *CP1* promotes the decay of CRU proteins. CRU protein decay is controlled through *CP1* maternal gametophytic alleles. (**A**) Endosperms from non-after-ripened (No aft.-rip.) WT and *cp1* seeds or from WT and *cp1* seeds after-ripened for short (three days) or long (two months) time periods were dissected at the indicated times after seed imbibition. Proteins isolated from the same number of WT (Col) or *cp1* endosperms were separated on SDS PAGE gels and probed with 12S antibody serum. The specificity of the antibody was confirmed using protein extracts isolated from the endosperm and embryo of the *cruabc* mutant (**Figure 4—figure supplement 1**). For a given after-ripening time, WT and *cp1* endosperm proteins were separated on the same SDS PAGE gel. An aspecific band is used as a loading control. Percent germination at each time-point after seed imbibition is indicated. Arrows indicate highly abundant cruciferin (CRU) proteins. (**B**) WTx*cp1* F1 and *cp1*xWT F1 seeds (obtained by reciprocally crossing WT (Col) and *cp1* plants) were subject to dry after-ripening treatments as in (**A**). For each after-ripening treatment, endosperms and embryos were dissected 36 hr after seed imbibition. Proteins isolated from the same number of endosperms or embryos were processed as in (**A**). Seeds obtained after crossing *cp1/+* heterozygous mother plants with Col WT (+/+) pollen were after-ripened for a short time period (three days). Endosperms were dissected 36 hr after

*Figure 4 continued on next page*

*Figure 4 continued*

imbibition, whereas embryos were further cultured for later genotyping in order to distinguish endosperms according to their (i) +/+/+ or (ii) *cp1/cp1/*+ genotype. Endosperms with the same genotype were pooled. Proteins extracted from the two endosperm pools were processed as in (**A**).

The following figure supplements are available for figure 4:

**Figure supplement 1.** Antibody specificity.

**Figure supplement 2.** *CP1* promotes CRU protein decay.

**Figure supplement 3.** *CP1* promotes CRU protein decay.

**Figure supplement 4.** *CP1* is not required to regulate germination sensu stricto.

**Figure supplement 5.** Embryo CRU protein levels are constant during early seed imbibition times.

**Figure supplement 6.** Procedure to assess CRU protein decay through maternal *CP1* alleles.

**Figure supplement 7.** Procedure for the assessment of CRU protein decay through *CP1* maternal gametophytic alleles.

**Figure supplement 8.** Monoallelic *CP1* expression rises upon imbibition in dormant seeds.

endosperm 36 hr after imbibition (*Figure 4B* and *Figure 4—figure supplement 2B*). Similar observations were made with WTx*cp1-3* and *cp1-3*xWT F1 seeds (*Figure 4—figure supplement 2B*).

Therefore, the *cp1* mutant maternal allele imposes a delay in CRU decay only in intermediately dormant seeds in which *CP1* preferential maternal allele expression could be detected in F1 seeds (*Figure 3*).

## Decay in CRU proteins is controlled through *CP1* maternal gametophytic alleles

Delayed CRU decay in *cp1*xWT F1 endosperm could reflect the *cp1* mutant genotype of the mother plant (a sporophytic maternal effect). To address this possibility, we used a *cp1/CP1* heterozygous mother plant in a cross with a WT pollen donor (*Figure 4—figure supplement 7*). This cross generates seeds whose endosperm is *CP1/CP1/CP1* or *cp1/cp1/CP1*, in which the *cp1* alleles originate from a *cp1* mutant female gametophyte (*Figure 4—figure supplement 7*). On the other hand, given that the sporophytic mother is *cp1/CP1* heterozygous, the maternal tissues bearing the seeds, including the seed testa, are genetically identical (*Figure 4—figure supplement 7*). *Figure 4B* and *Figure 4—figure supplement 2B* show that among seeds produced in the same *cp1/CP1* silique, the decay of CRUs was delayed in *cp1/cp1/CP1* endosperms relative to *CP1/CP1/CP1* endosperms 36 hr after seed imbibition.

These results therefore show that the maternally imposed delay in the decay of CRU proteins results, at least in part, from *cp1* mutant alleles present in the maternal gamete cells rather than in maternal sporophyte cells.

## *ALN* negatively regulates dormancy through maternal sporophytic and gametophytic alleles

The *ALN* gene encodes allantoin amidohydrolase (allantoinase). Allantoinase participates in the plant purine catabolism pathway by converting allantoin to allantoate, which is subsequently used for nitrogen remobilization. Besides its housekeeping function, there is increasing evidence that the purine degradation pathway participates in the plant's responses to biotic and abiotic stresses. In this context, Watanabe et al. showed that increased allantoin levels in response to drought stress protects the plant by enhancing ABA synthesis (*Watanabe et al., 2014*). Indeed, Watanabe et al. showed that *aln* seedlings accumulate high ABA levels (*Watanabe et al., 2014*). Given the importance of ABA in promoting seed dormancy, these results suggest that *ALN* could regulate seed dormancy. Indeed, we found that *aln* mutant seeds displayed high dormancy relative to WT seeds

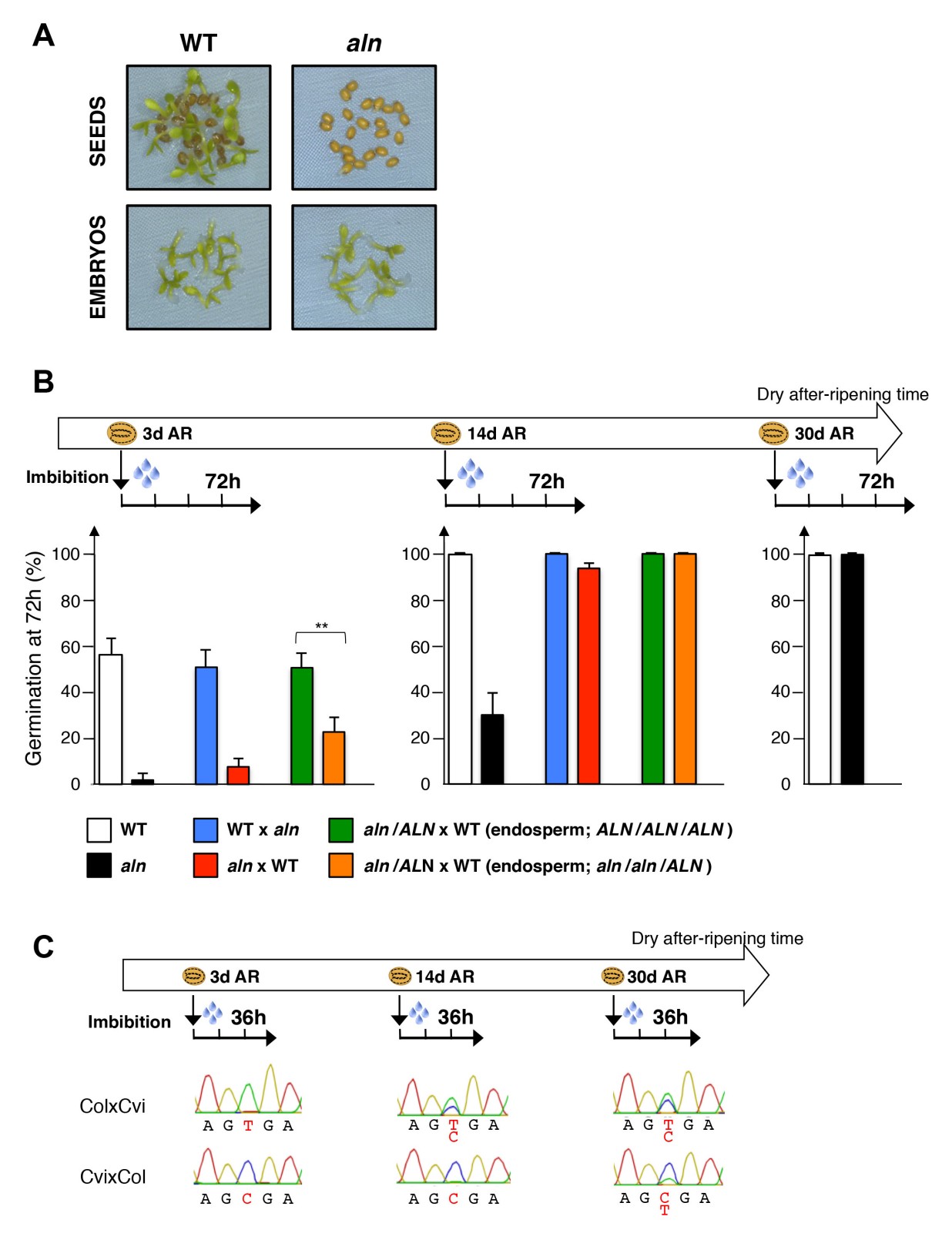

**Figure 5.** *ALN* affects seed dormancy level through maternal alleles. (**A**) WT and *aln* mutant seeds were after-ripened for 10 days. Pictures show seeds (upper panel) and embryos (lower panel) 72 hr after imbibition. Embryos were dissected 4 hr after imbibition. (**B**) WT, *aln*, WTx *aln* F1, *aln* x WT F1 and *aln*/*ALN* x WT F1 seeds were after-ripened for 3 days, 14 days, and 30 days (only for WT and *aln* seeds). Germination was scored 72 hr after seed imbibition (4 replicates, n = 50, **p<0.01, Student's t test). After germination, *aln*/*ALN* x WT individuals were genotyped to distinguish endosperms

*Figure 5 continued on next page*

Figure 5 continued

bearing *aln/aln/ALN* and *ALN/ALN/ALN* alleles. (C) Sanger sequencing chromatograms covering SNPs present in *ALN*. CvixCol and ColxCvi F1 seeds were dry after-ripened for 3 days, 14 days, and 30 days. RNA extracted from endosperm dissected 36 hr after imbibition was processed for RT-PCR followed by Sanger sequencing. Nucleotides at SNP sites are highlighted with red.

(*Figure 5A*). Thus, *ALN* is a negative regulator of seed dormancy. As expected, implementation of seed dormancy in *aln* mutants required the presence of the endosperm (*Figure 5A*).

As above, we next explored whether seed dormancy is maternally regulated by *ALN*. We generated *aln*xWT and WTx*aln* F1 seeds, developed in *aln* and WT mother plants, respectively. After a short after-ripening period of three days, the germination percentage of *aln*xWT F1 seeds was low and comparable to that of *aln* mutants (*Figure 5B*). By contrast, the germination percentage of WTx*aln* F1 seeds was markedly higher and similar to that of WT seeds (*Figure 5B*). The germination differences between *aln*xWT and WTx*aln* F1 seeds were no longer observed when the after-ripening time was prolonged to 14 days whereas *aln* mutant seed germination remained significantly lower than that of WT (*Figure 5B*). These results indicated the occurrence of maternal inheritance in seed dormancy levels.

Consistent with our results with *CP1*, preferential maternal allele expression of *ALN* in ColxCvi F1 seeds was detected in seeds after-ripened for three days but no longer detected after 14 days of after-ripening (*Figure 5C*). This suggested that maternal inheritance of seed dormancy levels could result from imprinted *ALN* gene expression. However, unlike *CP1*, *ALN* is expressed not only upon seed imbibition but also during embryogenesis (*Penfield et al., 2006*; *Winter et al., 2007*; *Bassel et al., 2008*; *Le et al., 2010*). To assess the relative contribution of sporophytic and gametophytic maternal allele effects on seed dormancy levels, we pollinated *aln/ALN* heterozygous plants with WT pollen. *Figure 5B* shows the average germination percentage of seeds bearing *ALN/ALN/ALN* or *aln/aln/ALN* endosperm. Although the differences in germination percentage were not as pronounced as those found with *aln*xWT and WTx*aln* F1 seeds, seeds bearing *aln/aln/ALN* endosperm were significantly more dormant than those bearing *ALN/ALN/ALN* endosperm. These results therefore strongly indicate that *ALN* negatively regulates dormancy at least in part by maternal gametophytic alleles.

## Discussion

Here we showed that hybrid seeds produced by Cvi and C24 reciprocal crosses or Cvi and Col reciprocal crosses tend to inherit dormancy levels more akin to those of the maternal ecotype (*Figure 1*). We also showed that the endosperm's germination-repressive activity is essential to implement maternally imposed and cold-induced dormancy levels (*Figure 2—figure supplement 6*). These observations further emphasize the key role played by the mature endosperm in implementing dormancy in *Arabidopsis*.

We identified a developmentally dynamic genomic imprinting expression program in the endosperm of mature seeds. Indeed, different sets of MEGs and PEGs were identified according to the dormant state of the seed. Concerning the inheritance of maternal dormancy levels observed in hybrid seeds, a substantial number of identified MEGs in dormant seeds are directly or plausibly involved in controlling seed germination and, furthermore, their expression correlates with seed dormancy levels (*Figure 2D* and *Figure 2—source data 3*).

More specifically, in shortly after-ripened seeds, *CP1* expression is low relative to fully after-ripened seeds and preferentially maternal. However, *CP1* expression is higher in ColxCvi seeds relative to that in CvixCol seeds, reflecting the lower dormancy of ColxCvi seeds, i.e. reflecting the lower dormancy of the maternal Col ecotype (*Figure 3* and *Figure 6B*). As dry seed after-ripening time proceeds, seeds progressively lose the capacity to sustain low *CP1* maternal allele expression. This marks the beginning of acquisition of the capacity of seeds to trigger the decay of CRU protein stores. This occurs faster in ColxCvi seeds than in CvixCol seeds (*Figure 6B*). How loss of preferential maternal allele expression is regulated remains unknown.

Furthermore, we identified *ALN* as a negative regulator of seed dormancy (*Figure 5*). *ALN* is a dormancy-specific MEG whose expression, like that of *CP1*, negatively correlates with dormancy

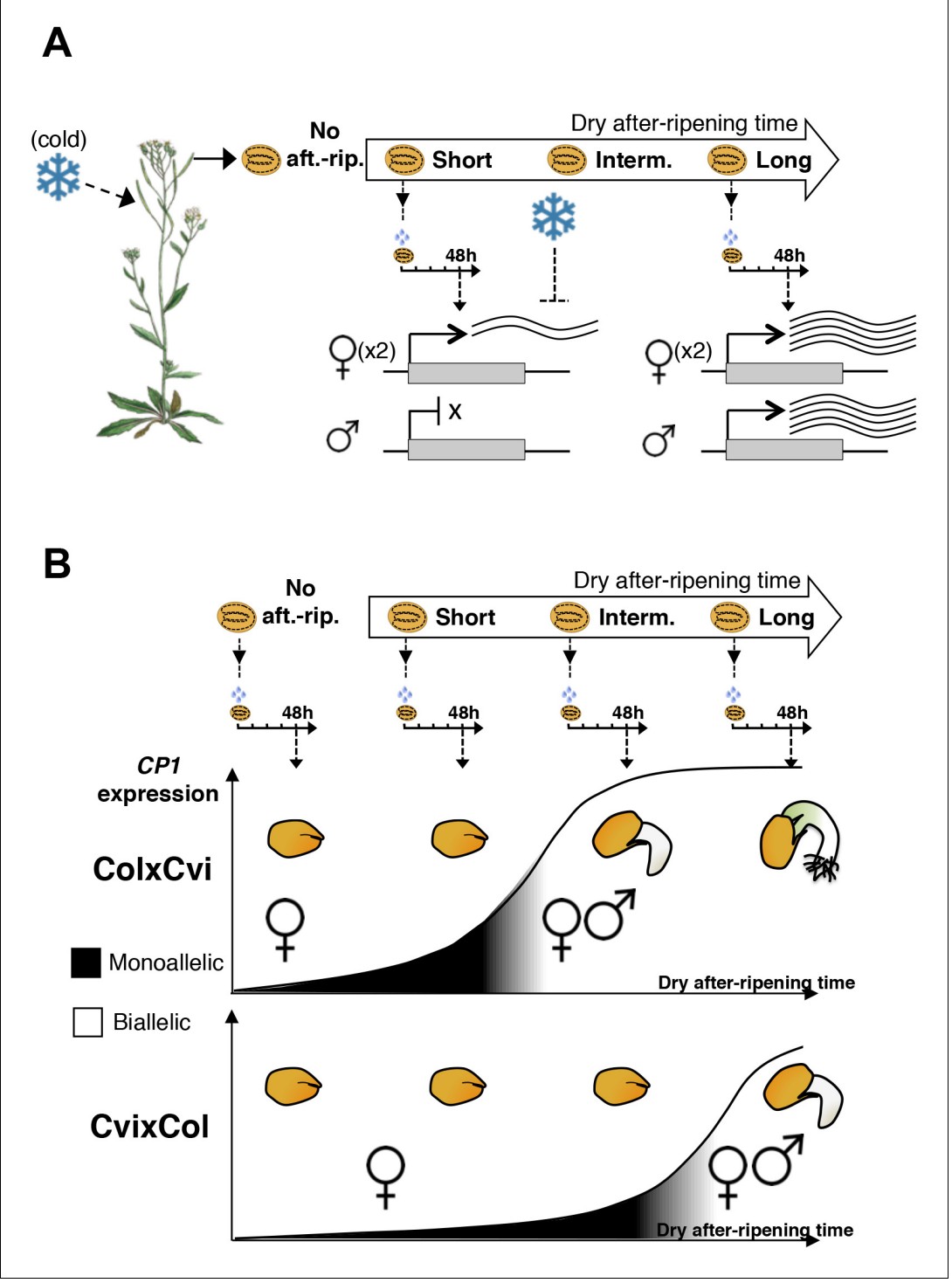

**Figure 6.** A model for maternal dormancy inheritance in seed hybrids. (**A**) General mechanism accounting for maternal inheritance of seed dormancy in hybrid seeds. Upon imbibition, a group of genes that regulate seed germination processes are preferentially maternally expressed during the seed-dormant period. As a result, dormancy levels in hybrid seeds reflect the maternal genotype. (**B**) The case of *CP1*. In the absence of after-ripening (No aft.-rip), preferential maternal allele expression is very low. As seeds begin to after-ripen (Short), preferential maternal allele expression rises according to the maternal genotype, i.e. there is higher maternal allelic expression in ColxCvi F1 relative to CvixCol F1 seeds (see also *Figure 3*). After an intermediate after-ripening period, low and maternal allelic *CP1* expression is lost in ColxCvi F1 seeds but not in CvixCol F1 seeds. As a result, seed CRUs decay is initiated earlier in ColxCvi F1 seeds relative to CvixCol F1 seeds. Further after-

*Figure 6 continued on next page*

*Figure 6 continued*

ripening abolishes preferential maternal allelic expression in CvixCol F1 seeds upon imbibition, which leads to undelayed decay of CRU proteins and seed germination.

levels. We provide direct evidence that it regulates dormancy, at least partially through maternal gametophytic alleles (*Figure 5*).

Together, these observations suggest a model in which maternal dormancy inheritance involves the preferential maternal allele expression of regulators of seed dormancy upon seed imbibition. Therefore, dormancy levels in hybrid seeds reflect the maternal genotype, reflected in the maternal expression levels or product activities of genes regulating dormancy (*Figure 6*).

The occurrence of genomic imprinting upon seed imbibition raises the question of which developmental stage, i.e. prior or after mature seed formation, is the one in which maternal allelic transcription takes place. It appears unlikely that this transcription takes place only during seed development, with resulting mRNAs being stored in mature seeds. Indeed, we found that *CP1* maternal allelic mRNA accumulation increased over time after imbibition, indicating de novo and dormancy-specific maternal allele transcription (*Figure 4—figure supplement 8*). Furthermore, the MEGs and PEGs identified in non-dormant seeds probably involve maternal allelic transcription upon seed imbibition as they were not detected in dormant seeds. These observations support the notion of a dedicated parental-specific allele transcription program operating upon mature seed imbibition.

The biological significance of maternally inherited dormancy remains to be understood. Presumably, the mother plant adjusts the dormancy of its seed progeny to optimize future germination according to seasonal cues. Interestingly, the majority of dormancy-specific MEGs whose expression correlates with dormancy levels also have their expression further regulated by cold during seed development. This suggests that imprinted gene expression levels upon imbibition are further adapted to the environmental cues perceived by the mother plant or by the developing seed tissues or both.

The evolutionary driving force leading to imprinting in the context of seed dormancy is unclear. Although the following scenario might seem unlikely for *Arabidopsis*, which is mainly a self-fertilizing species, it might reflect an evolutionary trend present in related cross-pollinating species. Indeed, in the case of a seed fathered by a pollen grain coming from a distant plant, which could be either a different accession or have experienced different climatic conditions than the mother plant, the paternal allele expression may interfere with maternal control of seed dormancy. It would seem advantageous therefore to silence paternal genes that regulate germination to prevent such paternal interference. Consistent with this speculation, the number of PEGs identified in dormant seeds is markedly lower than that reported previously during early embryogenesis (*Figure 2*) (*Pignatta et al., 2014*).

In plants, genomic imprinting during early embryogenesis is mainly proposed to arise from a 'parental conflict' over food allocation to the offspring (*Jiang and Köhler, 2012*; *Köhler et al., 2012*; *Gehring, 2013*; *Rodrigues and Zilberman, 2015*). The occurrence of dormancy-specific genes would rather seem to reflect a 'mutual interest' between progenitors rather than a 'conflict' (*Haig, 2014*).

## Material and methods

### Plant material

The *Arabidopsis thaliana cp1* and *aln* mutants were obtained from The European Arabidopsis Stock Centre (RRID:SCR_004576). Salk_067293 (*cp1-3* allele) bears a T-DNA insertion in the promoter region of *CP1*. salk_146500 (*cp1-1 allele*) bears a T-DNA insertion in the second exon of *CP1* and salk_020878 (*cp1-2* allele) in the second intron of *CP1*. Salk_109258 (*aln*) bears a T-DNA insertion in the promoter region of *ALN.* Cvi tetraploid and C24 tetraploid seeds were kindly provided by Dr. Luca Comai and *cruabc* triple mutant was kindly provided by Dr. Leonie Bentsink and described by *Withana-Gamage et al. (2013)*.

## Plant growth conditions and germination assays

When comparing hybrid seed germination or gene expression, hybrid seed material was obtained after crossing parents on the same day. Dry siliques were obtained about three weeks after pollination. Seeds from dry siliques were harvested on the same day from plants grown side by side under identical environmental conditions (22–24°C, 100 µE/m$^2$/s, 16 hr/8 hr day/night photoperiod, 70% relative humidity). Seeds were dry after-ripened at room temperature (22–24°C, 50–60% relative humidity) for indicated time periods. For each individual experiment, seeds were produced from plants grown under the exact same environmental conditions. Nevertheless, different seed batches may contain variations in dormancy levels because the amount of after-ripening time needed to break dormancy can vary. For the Col ecotype, variation is in the order of 1–5 days, whereas for the Cvi ecotype, variations can be up to several weeks.

For each genotype, germination assays were performed with seeds from at least four independent siliques (30–50 seeds in each silique). For germination tests, seeds were sterilized (1/3 bleach, 2/3 water, 0.05% Tween) and plated on a Murashige and Skoog medium containing 0.8% (w/v) Bacto-Agar (Applichem).

Plates were incubated at 21–23°C, 16 hr/8 hr day/night photoperiod, light intensity of 80 µE/m$^2$/s, humidity of 70%. Between 100 and 200 seeds were examined with a Stemi 2000 (Zeiss) stereomicroscope and photographed with a high-resolution digital camera (Axiocam Zeiss) at different times after seed imbibition. Photographs were enlarged electronically for measurement of germination events (i.e. endosperm rupture events). Percent of germination of C24xCvi, CvixC24, C24xCvi$^{tet}$ and CvixC24$^{tet}$ F1 seed populations was scored in three independent experiments giving similar results.

For dissected embryo growth analysis, seeds of each genotype were sterilized and plated under standard germination conditions. Four hours after seed imbibition, dissected embryos were cultivated for four days under standard germination conditions. Pictures were taken four days after embryos dissection.

## Statistics

Errors bars in histograms correspond to SD values. We used Student's t-test (two-tailed assuming unequal variance) to compare average mean values in order to determine whether their differences were statistically significant (** $p<0.01$, ***$p<0.001$).

## RNA extraction and RT-PCR/ RT-qPCR

Total RNA was extracted as described by Vicient and Delseny (1999). When extracting RNA from 40 partially dissected endosperm, 40 fully dissected endosperms and 40 dissected testas, the Pellet Paint Co-precipitant (Merck, Switzerland) was used for final RNA precipitation. Total RNAs were treated with RQ1 RNase-Free DNase (Promega, Switzerland) and reverse-transcribed using ImpromII reverse transcriptase (Promega) and oligo(dT)15 primer (Promega) according to the manufacturer's recommendations. Quantitative RT–PCR was performed using the ABI 7900HT fast real-time PCR system (Applied Biosystems, Switzerland) and Power SYBR Green PCR master mix (Applied Biosystems). Relative transcript levels were calculated using the comparative ΔCt method and normalized to the *PP2A* (*AT1G69960*) gene transcript levels. Primers used in this study are listed in *Supplementary file 2*.

## RNAseq

Seeds were sterilized and plated under standard germination conditions (*Piskurewicz and Lopez-Molina, 2016*). Dissection procedures were performed on WT seeds (Col-0) 36 hr after imbibition. Total RNA was extracted as described before (*Piskurewicz and Lopez-Molina, 2011*) and RNA concentrations were measured by Qubit Fluorometric quantification system (Thermo Fisher Scientific, Switzerland). For partially dissected endosperm and fully dissected endosperm samples, the cDNA libraries were prepared from 200 ng total RNA using a TruSeq mRNA Library Prep Kit (Illumina, Switzerland). Although testa RNA could not be detected, we prepared the cDNA library with the same volumes of RNA testa as those used for the construction of the endosperm cDNA libraries. cDNA libraries were normalized and pooled then sequenced using HiSeq 2500 (Illumina) with single-end 100 bp reads. For sequencing testa cDNA, we used the same volume of cDNA as that used for sequencing endosperm cDNA. cDNA library preparation and sequencing, as well as

read mapping and counting, were performed in the same manner for all seed materials (testa, partially and fully dissected endosperm). The resulting RNA concentrations, library concentrations, and read numbers are shown in *Table 1*.

## Identification of Cvi SNPs

We sequenced cDNA libraries prepared from endospermic RNA (partially dissected endosperm from short- and long-period after-ripened seeds). Low-quality reads were filtered out by the fastqc program (RRID:SCR_005539) with the option –q 20 and –p 90. The remaining reads were mapped to the Col-0 genome (TAIR10) with the TopHat program (RRID:SCR_013035), allowing up to five mismatches per alignment. The resulting two alignment files (BAM files) were merged into one file. Variant calling was performed using the FreeBayes program (RRID:SCR_010761) using the input filters option (-F –min-alternate-fraction 0.9 and -C –min-alternate-count 3) and the resulting SNPs were compared to the publicly available Cvi SNPs (http://www.arabidopsis.org/download_files/Polymorphisms_and_Phenotypes/Ecker_Cvi_snps.txt). This led us to identify 107,977 common exonic SNPs covering 15,814 genes, which in turn were used to distinguish Cvi and Col reads in ColxCvi and CvixCol F1 RNA sequencing data. The SNPs used in this study are listed in *Supplementary file 3*.

## Identification of imprinted genes

We sequenced cDNA libraries prepared from ColxCvi and CvixCol endospermic F1 RNA. Low-quality reads were filtered out and the remaining reads were treated as described above for the Cvi reads.

We called variants at each SNP position using the SAMtools Mpileup (RRID:SCR_002105) supplied with a reference bed file of SNP positions to generate a pileup file. The resulting file was used with the SAMtools Filter pileup program, which generated read counts for the reference (Col) and variant (Cvi) genome for every SNP position. Next, reads specific to Col and Cvi alleles were summed across all the SNPs present in each gene. The resulting numbers for each gene are provided in *Figure 2—source data 6*. A list of potential maternally expressed genes (MEGs) and paternally expressed genes (PEGs) was established after applying the following criteria: (1) potential MEGs are those genes whose ratio of maternal allele to paternal allele read number is more than 0.8 in both ColxCvi and CvixCol endosperm F1 material; similarly potential PEGs are those whose ratio of paternal to maternal allele read number is more than 0.6; (2) potential MEGs or PEGs with read numbers lower than five were discarded; (3) the probabilities of deviation from expected ratio (maternal:paternal = 2:1) in a Fisher's two-tailed exact test were calculated and only potential MEGs or PEGs with a $p < 0.05$ were retained.

## Validation of imprinting

RNA was extracted from other sets of reciprocal crosses between Col and Cvi (after ripening for five days and two months), treated with DNase, and cDNA was synthesized using ImpromII reverse transcriptase (Promega) and oligo dT primers. We selected 17 MEGs and two PEGs and designed primers to amplify cDNA fragments containing SNPs. PCR amplicons were purified using Wizard SV Gel and PCR Clean-Up System (Promega), then sequenced by Sanger sequencing. The same PCR amplicons were pooled and libraries were constructed using the Illumina NexteraXT kit (Illumina). Libraries were sequenced using the MiSeq system (Illumina) with paired-end 150 bp reads. The resulting reads were mapped to the Col-0 genome (TAIR10) using the TopHat software (RRID:SCR_013035). Col and Cvi reads at the SNP position of each gene were counted as described above for RNA-seq.

## Gene expression profile

Transcripts assembly and normalization was performed with the Cufflinks program (RRID:SCR_013307), and gene expression levels were calculated in FPKM (Fragments Per Kilobase of exon per Million mapped fragments) units. Differential gene expression analysis was performed by Cuffdiff (RRID:SCR_001647), a part of the Cufflinks package. For clustering analysis, we used relative expression data compared with the ColxCvi sample. The rest of the genes were clustered using K-means clustering in R statistical computing. Mapping reads, SNP calling, and gene expression analysis were

performed using Galaxy (RRID:SCR_006281) (*Giardine et al., 2005*; *Blankenberg et al., 2010*; *Goecks et al., 2010*).

## Confocal microscopy

The marker lines of *pFWA::dFWA-GFP* (*Kinoshita et al., 2004*) and *pDD65::mtKaede* (mtKaede: mitochondria localized Kaede-GFP protein) (*Arimura et al., 2004*) were crossed reciprocally. The F1 seeds were surface-sterilized and plated on the 1/2 MS-Agar media, incubated under 16 hr/8 hr of light and dark photoperiod at 22°C for 36 hr. The endosperm was dissected under a binocular microscope. The fluorescent signals from the endosperm cell layer were captured using the confocal laser microscope (Olympus FV-1000) equipped with 488 nm laser, a x40 objective lens (UPLSAPO 40X, WD = 0.18, NA = 0.95; Olympus) and band-pass filter cube of 520/35 nm for GFP. The images were were adjusted for brightness and contrast using Adobe Photoshop CS6 (Adobe systems, Inc).

## Protein gel staining and western blot analysis

Proteins were extracted from 30 endosperms or three embryos using Laemmli buffer (100 mM Tris pH 6.8, 200 mM DTT, 4% SDS, 20% glycerol, 0.02% bromophenol blue). Proteins extracted from 30 endosperms or from three embryos (i.e. 3 μg of total protein) were resolved under reducing conditions using 10% SDS/polyacrylamide gels. Gels were stained with a solution of 0.25% coomassie brilliant blue R-250 (Sigma) in 40% methanol +10% acetic acid. For western blot analysis, proteins were extracted as described above. The amount of protein loaded in each line of the gels is an equivalent of two endosperms (i.e. 0.2 μg of total protein) or 0.5 embryos (0.5 μg of total protein). Proteins were resolved under reducing conditions using 12% SDS/polyacrylamide gels, transferred to polyvinylidene fluoride membrane and probed with 12S antibody serum as described by *Barthole et al. (2014)*. Specificity of the antibody was confirmed using protein extracts isolated from the endosperm and embryo of *cruabc* mutants (*Figure 4—figure supplement 1*).

## Gametophytic effect of maternal *cp1* inheritance

A heterozygous *cp1/CP1* mother plant was pollinated with wildtype (*CP1/CP1*) Col-0 pollen (*Figure 4—figure supplement 5*). The resulting mature siliques were harvested on the same day and seeds were after-ripened for 2 days. Seeds were sterilized and plated under standard germination conditions. 36 hr after imbibition, individual seeds were dissected and partially dissected endosperm material was suspended in 1 μl of SDS-PAGE loading buffer and frozen. The embryos were further cultured for two weeks prior to genotyping to identify and distinguish embryos carrying a *cp1-1* allele from those carrying a *CP1* allele. Thirty *cp1/cp1/CP1* partially dissected endosperms (coming from seeds with *cp1/CP1* embryos) and 30 *CP1/CP1/CP1* partially dissected endosperms (coming from seeds with *CP1/CP1* embryos) were separately pooled. 20 μl of partially dissected endosperm protein extracts from each pool were run side-by-side on an SDS PAGE gel and probed with the 12S antibody serum or stained with coomassie blue solution.

## Acknowledgements

We thank Dr. Luca Comai for kindly providing Cvi and C24 tetraploid seeds. We thank Leónie Bentsink for providing *crua/crub/cruc* mutant seeds. We thank Colette Larre for providing antibodies against CRUs. We thank Mylène Docquier and members of the Genomics Platform of the Institute of Genetics and Genomics (iGE3) at the University of Geneva for their invaluable help in conducting sequencing experiments. Grant-in-Aid for Scientific Research on Innovative Areas Nos. 16 H06465, 16 H06464, and 16 K21727 to TK. Work in LLM's laboratory was supported by grants from the Swiss National Science Foundation and by the State of Geneva.

## Additional information

### Funding

| Funder | Grant reference number | Author |
| --- | --- | --- |
| Schweizerischer Nationalfonds zur Förderung der Wis- | 31003A_152660 | Urszula Piskurewicz Mayumi Iwasaki |

| | | |
|---|---|---|
| senschaftlichen Forschung | | Christian Megies<br>Luis Lopez-Molina |
| Japan Society for the Promotion of Science | Grant in Aid for Scientific Research on Innovative Area 16H06465 | Daichi Susaki<br>Tetsu Kinoshita |
| Japan Society for the Promotion of Science | Grant in Aid for Scientific Research on Innovative Area 16H06464 | Daichi Susaki<br>Tetsu Kinoshita |
| Japan Society for the Promotion of Science | Grant in Aid for Scientific Research on Innovative Area 16K21727 | Daichi Susaki<br>Tetsu Kinoshita |

The funders had no role in study design, data collection and interpretation, or the decision to submit the work for publication.

## Author contributions

UP, MI, LL-M, Conception and design, Acquisition of data, Analysis and interpretation of data, Drafting or revising the article, Contributed unpublished essential data or reagents; DS, CM, TK, Acquisition of data, Analysis and interpretation of data, Contributed unpublished essential data or reagents

## Author ORCIDs

Urszula Piskurewicz, http://orcid.org/0000-0002-9523-9123
Luis Lopez-Molina, http://orcid.org/0000-0003-0463-1187

## Additional files

### Supplementary files

• Supplementary file 1. List of germination-related MEGs.

• Supplementary file 2. Primers used in this study.

• Supplementary file 3. Cvi SNPs data.

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
