## [Decision Letter]

Thank you for submitting your article "Dormancy-specific imprinting underlies maternal inheritance of seed dormancy in *Arabidopsis thaliana*" for consideration by *eLife*. Your article has been reviewed by four peer reviewers, one of whom is a member of our Board of Reviewing Editors, and the evaluation has been overseen by Detlef Weigel as the Senior Editor. The following individual involved in review of your submission has agreed to reveal his identity: Steven Penfield (Reviewer #4).

The reviewers have discussed the reviews with one another and the Reviewing Editor has drafted this decision to help you prepare a revised submission.

This paper reports a transient imprinting phenomenon in mature *Arabidopsis* endosperm that appears to involve a different set of genes from those identified in earlier development. The reviewers felt that this is an interesting and important study that presents novel and exiting findings. The paper suggests an unappreciated new role for imprinting, shows for the first time that imprinting persists into maturity, and that imprinting is dynamic with respect to dormancy state and environmental temperature during seed maturation.

However, there are four key areas in which substantial revisions are required:

1) Verification of the new imprinting phenomenon.

The report of gene imprinting in mature seeds is novel and exciting, but there are several reasons to be cautious about the findings. a) The imprinted genes reported are almost completely different from known imprinted genes. b) There is little overlap between genes imprinted in dormant and non-dormant seeds. c) The imprinting phenomenon is transient. d) Only one replicate for each cross was used. e) Some of the Sanger sequencing chromatograms used for validation are inconsistent. For instance, in Figure 2, ABCG30 in non-dormant seeds seems mono-allelic and VIM5 in non-dormant seeds seems bi-allelic, unlike stated. This kind of inconsistency is also observed in several cases in Figure 2—figure supplement 2. f) The p-value used to call imprinted genes (0.05) is quite permissive, and likely led to inclusion of false positives. g) The RNA-seq results are not always consistent between reciprocal crosses. For example, CP1 has 5683 maternal vs. 17 paternal reads in Col x Cvi, but only 56 maternal vs. 10 paternal reads in Cvi x Col. h) Only MEGs specific for the dormant dataset were tested by Sanger sequencing.

For these reasons, it is imperative that the imprinting results are thoroughly verified using a robust, quantitative technique. We suggest RT-PCR of multiple genes from both the dormant and non-dormant datasets, using RNA samples different from those used for RNA-seq, followed by sequencing on an Illumina MiSeq instrument (many PCR products can be multiplexed in one lane). Furthermore, we think it is important to verify parent-of-origin specific expression in mature endosperm using a reporter construct. Because creating a new reporter construct would take a long time, we suggest using the published FWA:GFP reporter.

In addition to new experiments, please provide more information about the analysis of imprinted genes. Specifically, please include in a supplementary table parent-of-origin RNA-seq counts for all genes, not just those deemed significantly imprinted, and please state in the text whether most genes conform to the expected 2:1 maternal/paternal ratio. Also, please explain in more detail in the methods section how imprinted genes were called, particularly how information from the reciprocal crosses was used.

2) A direct functional link between imprinted expression and dormancy.

The reviewers are in agreement that the paper presents evidence for a link between imprinting and dormancy, but not direct evidence that imprinted genes regulate dormancy. The paper shows that maternal control of dormancy cannot be explained by the predominance of maternal genetic material in the endosperm, but this is not proof that imprinted expression controls dormancy. Signals from the maternal seed coat could influence the state of the endosperm, so that even after the seed coat degenerates, how endosperm regulates dormancy may be governed by earlier seed coat signals. Correlation of imprinted gene expression with dormancy can run in either direction: gene expression may influence dormancy, or dormancy can influence gene expression. The gene that was examined in detail, CP1, fits the latter pattern. The effects of the *aln* mutation were not tested in F1 seeds derived from reciprocal crosses with wild-type, so it remains unclear whether maternal expression of ALN regulates dormancy. Therefore, the title of the paper presently does not accurately reflect the reported findings.

We feel the best way to remedy this would be to show that maternal expression of ALN (or another gene, such as KAI2) regulates dormancy. Depending on the results of these experiments, the paper's conclusions may need to be adjusted to reflect a link with dormancy instead of a causative relationship.

3) Quantification of cruciferin (CRU) proteins.

There is concern that the quantification of CRU proteins reported in this paper may not be reliable. First, the loading control for the CRU assays may be inadequate. In Figure 4 another protein band is used as a loading control, but when looking at Figure 4, this band varies greatly between samples. This raises the question of whether there is indeed a depletion of the putative CRUs during dormancy break, or if the low band intensity is due to less total protein being loaded on the gel. Second, reliance on coomassie-stained gels, instead of Western blots, to determine CRU levels may lead to inaccurate CRU quantification. To address these concerns, please include an adequate loading control, and either utilize Western blots or explain why coomassie staining is a reliable method.

4) Manuscript readability.

During initial editorial assessment of the manuscript and peer review, a common concern was that the writing presents difficulties even for experts, with descriptions like "circular", "turgid" and "repetitive". Please do your best to improve the logical flow and accessibility of the paper, keeping in mind the broad readership of *eLife*. To assist you, we are enclosing the original reviews below, which include many suggestions for manuscript improvement.

Reviewer #1:

This paper reports a transient imprinting phenomenon in mature *Arabidopsis* endosperm that appears to involve a different set of genes from those identified in earlier development. This imprinting phenomenon doesn't obviously fit the established mechanisms, where most MEGs are directly activated by DNA demethylation (which shouldn't be transient), and PcG regulation mostly occurs at PEGs. The authors show that the expression of many of these genes correlates with dormancy, and show that maternal expression of the protease CP1 regulates the levels of cruciferin storage proteins.

Overall, I think this is an interesting manuscript that makes two important contributions: the discovery of a new set of imprinted genes, and the strong implication that some of these genes are involved in seed dormancy – a function that doesn't easily fit the predominant parental conflict hypothesis. Considering how little we know about the functions of plant imprinted genes, this paper presents important progress.

This said, this manuscript has substantial weaknesses. The greatest is the absence of unambiguous evidence that imprinting regulates dormancy. The authors show that maternal control of dormancy cannot be explained by the predominance of maternal genetic material in the endosperm, but this is not proof that imprinted expression controls dormancy. Signals from the maternal seed coat could influence the state of the endosperm, so that even after the seed coat degenerates, how endosperm regulates dormancy may be governed by earlier seed coat signals. Correlation of imprinted gene expression with dormancy can run in either direction: gene expression may influence dormancy, or dormancy can influence gene expression. The gene that the authors chose to examine in detail, CP1, fits the latter patterns. The authors state that several of the imprinted genes are known regulators of dormancy – why didn't they test whether maternal expression in the endosperm of one or more of these is important for dormancy? I think this would be the single most important addition to the paper, which would clearly establish that imprinted expression of a particular gene or genes regulates dormancy.

The other major weakness, related to the above, is that the manuscript makes for a difficult read. The text is frequently dense, circuitous and repetitive, and the logical thread from dormancy to CP1 is hard to follow.

A smaller issue is the lack of any mechanistic insight into how the newly identified genes are imprinted. This is probably beyond the scope of this manuscript, but the authors could at least check if the new imprinted genes are enriched in DME-catalyzed DMRs or PcG targets (as measured by H3K27me3).

Reviewer #2:

In this work, Piskurewicz et al. report that maternal regulation of seed dormancy in *Arabidopsis* may be caused by the differential expression of maternal and paternal alleles in the endosperm (imprinted genes). This finding is novel and exciting; however, I have several concerns that require to be addressed to fully support this major claim.

1) One major concern with this work is the lack of details for the analysis of imprinted genes. Apparently no replicates have been generated, limiting the value of the analysis. It is also not clear how many genes follow the expected ratio of two maternal to one paternal and how many deviate. Furthermore, how many genes are commonly imprinted in reciprocal crosses and how many are not? The data provided in [Supplementary-material SD6-data] do not allow to extract this information, partly because the labelling of the columns is incomplete, partly because only selected genes are shown.

2) To provide convincing proof that genes are specifically maternally expressed in the endosperm after imbibition, reporter genes should be tested. There are published reporters for e.g. FWA that could be used for this analysis.

3) The authors do not provide genetic evidence that the identified imprinted genes have a role in seed dormancy. Even if there is indeed a cluster of MEGs whose expression correlates with the dormancy status of different *Arabidopsis* accessions (and their hybrids), it is not possible to distinguish what is cause and consequence. It is possible that those genes are expressed because dormancy was broken rather than their expression being the cause for the break in dormancy. For instance, the authors propose that the differential expression of CP1, a MEG, may be involved in dormancy due to its potential role in protein storage decay. However, *cp1* mutant seeds do not seem to have different dormancy levels compared to WT. Therefore, while there may be a correlation between the expression of certain MEGs and dormancy levels, there is no evidence in this work for a functional link between the two.

4) I also have major concerns regarding the analysis of cruciferin (CRU) levels in this manuscript. Unlike stated by the authors, Barthole et al. (2014) do not rely on coomassie-stained gels to determine CRU levels, but instead make use of anti-CRU antibodies in a Western-blot. This raises the question of how can the authors in this manuscript distinguish CRUs from other abundant proteins? Furthermore, there is no adequate loading control for the assays. In Figure 4 the authors make use of another protein band for loading control, but when looking at Figure 4, this band varies greatly between samples. This raises the question of whether there is indeed a depletion of the putative CRUs during dormancy break, or if the low band intensity is due to less total protein being loaded on the gel. In my view, these two points raise major questions on whether the CRU-related data is reliable.

5) The chromatograms for the Sanger sequencing are sometimes dubious. For instance, in Figure 2, ABCG30 in non-dormant seeds seems mono-allelic and VIM5 in non-dormant seeds seems bi-allelic, unlike stated. This kind of inconsistency is also observed in several cases in Figure 2—figure supplement 2.

6) Why were only MEGs specific for the dormant dataset tested by Sanger sequencing?

7) The effect of *aln* on dormancy should be tested in F1 seeds derived from reciprocal crosses with wild-type to test whether the mutant phenotype has a maternal origin.

Reviewer #3:

Recently harvested seeds are often dormant and fail to germinate when they are imbibed. It is well-known that the maternal parent plays a major role in establishing dormancy, and that mature endosperm, a single layer of cells between the embryo and maternal testa cell layers, is essential for dormancy. Different *Arabidopsis* accessions display different levels of seed dormancy. C24 has low dormancy and Cvi has high dormancy, and the differences are quite dramatic. After a short 5-day after-ripening period, neither C24 or Cvi seeds germinate after 72 hrs imbibition. After an intermediate after-ripening period (25 days), full germination of C24 occurs, while almost no Cvi seeds germinate. Both C24 seeds and Cvi seeds after a long after-ripening period (6 months). The authors used these dormancy differences between C24 and Cvi to show that:

The maternal parent determines the level of seed dormancy. That is, F1 seed from C24xCvi have low seed dormancy to C24, and F1 seed from CvixC24 have high seed dormancy equivalent to Cvi.

The maternal bias is not due to the endosperm's 2:1 ratio of maternal:paternal chromosomes. That is F1 progeny C24xCvitetraploid and CvixC24tetraploid had similar levels of dormancy as F1 progeny between diploid parents.

These results motivated them to determine whether gene imprinting might underlie the maternal inheritance of seed dormancy.

Comment. Throughout the manuscript, the authors use the word "gene imprinting". However, the issue of whether a gene imprint exists that distinguish maternal versus paternal alleles is never addressed. Therefore, although it is OK to use the phrase gene imprinting, I think the phrase parent-of-origin gene expression should also be incorporated into the paper.

The authors convincingly show that the seed coat (endosperm + testa) can be used to isolate endosperm RNA that will be used to identify parent-of-origin gene expression in the endosperm because the testa contributes little mRNA.

For the high-throughput RNA-seq experiments, the authors replaced C24 with Col, which is considered to be a low dormancy accession. Following a short after-ripening period of 10 days, 40% of ColxCvi F1 seeds germinated whereas only 1% of CvixCol F1 seeds germinated. (Figure 1). When intermediately after-ripened for 20 days, about 80% of ColxCvi F1 seeds germinated whereas only 10% of CvixCol F1 seeds germinated.

Comment. In the Material and Methods section –, the authors state, "ColxCvi and CvixCol seeds were harvested the same day and after-ripened for 10 days (dormant seeds) or for 2 months (non-dormant seeds). Total RNA samples were isolated from seed coats (n=200) dissected 36 hours after seed imbibition."

This sentence should be moved into the body of the text because it clearly defines what is meant by "dormant seeds" and "non-dormant" seeds, terminology that is first used in subsection “Dormancy-specific genomic imprinting in the endosperm” paragraph seven for "dormant" CvixCol and ColxCvi F1 seeds, and "non dormant" CvixCol and ColxCvi F1 seeds.

It should be noted that after 10 days of ripening, 40% of the ColxCvi F1 seeds germinated and 60% did not, so they are a mixture of approximately equal numbers of dormant and non-dormant seeds. Perhaps the authors could explain why this time point was chosen.

Although it is likely that at 2 months all ColxCvi and CvixCol F1 seeds are non-dormant, the authors never show data supporting this, and only present the 6-month after-ripening data (see above).

The authors identified candidate imprinted genes by high-throughput sequencing RNA from F1 hybrid seeds; using SNPs to distinguish parent-of-origin expression.

They identified 67 MEGs/4 PEGs in dormant seed coat (endosperm) and 49 MEGs/8 PEGs for non-dormant seed coat (endosperm). Only 14 MEGs were present in both dormant and non-dormant seed coat. Only 5 MEGs and 3 PEGs have been previously identified in studies using seeds at earlier stages of development.

9 of the MEGs were selected for further analysis. Sanger sequencing experiments using F1 seed from reciprocally crossed parents was used to verify parent-of-origin expression. Interestingly, the parent-of-origin specific expression was dynamic – with many showing parent-of-origin expression at the dormant stage (10-day after-ripening) and biallelic expression at the non-dormant stage (2-month after-ripening).

As shown in Table 3, many of the dormancy-specific MEGs likely play roles in regulating dormancy/germination. The authors focused on the function of two particularly interesting MEGS – ALN and CP1.

ALN. As described in Watanabe et al. (2014, Plant Cell Environment, 37:1022) the purine metabolite, allantoin, promotes abscisic acid (ABA) production by activating transcription of a key enzyme for ABA biosynthesis, and by a post-translational activation mechanism. ABA is a major plant hormone that promotes dormancy. The ALN gene encodes allantoin amidohydrolase, which, according to Werner et al. (Plant Physiol 2008, 146:418) degrades allantoin. Consistent with this, *aln* loss-of-function mutations result in increased allantoin production, and increased ABA production.

Comment. I believe that the function of the ALN gene can be more clearly described in the manuscript. Also, only the Watanabe reference is cited, but it is not included in the references.

Importantly, the authors show that the *aln* mutant displays high dormancy relative to wild-type seeds. Thus, the ALN MEG gene is a positive regulator of dormancy.

The authors next raise the issue whether there is an ecotype bias for expression of the MEGs.

Comment. Analysis of expression after reciprocal crosses seemed to point to expression of alleles being controlled by their parent-of-origin, not by their respective ecotypes. So, it seemed that the issue of ecotype bias was already ruled out for gene imprinting.

However, the authors investigated ecotype-biased expression by assessing whether dormancy-specific MEG expression correlates with dormancy levels imposed by a given ecotype. They show that somewhat less than half (23 out of 53) had expression levels that negatively correlate with seed dormancy levels. That is, lower expression is correlated with higher dormancy.

Comment. I believe that the authors are not addressing an ecotype effect that affects the silencing of the paternal allele (gene imprinting). Rather, their clustering results point to a group of genes that conform to the ALN gene model, where lower maternal expression results in higher dormancy (see above).

CP1.

The CP1 gene encodes cysteine protease1, which promotes storage protein decay. CP1 expression is activated upon imbibition, resulting in release of amino acids from storage proteins to be used by the young seedling. CP1 is a MEG whose expression level increases with after-ripening time, but the level of CP1 expression is lower in Cvi (high dormancy) than in Col (high dormancy). Thus, CP1 expression, like ANL and the gene cluster (see above) is negatively correlated with dormancy.

The authors go on to show that the CP1 allele inherited from the maternal parent, and more specifically, the CP1 allele inherited from the maternal gametophyte, promotes decay of *Arabidopsis* seed storage proteins, cruciferins (CRU).

Finally, expression of CP1, like ANL, becomes biallelic as ripening time passes. Thus, the parent-of-origin expression is dynamic.

Discussion.

The authors state, "Concerning the MEGs and PEGs identified in non-dormant seeds it seems likely that their monoallelic transcription takes place upon seed imbibition, since they were not identified in dormant seeds. It appears therefore that a dedicated developmental program of genomic imprinting is operating in mature seeds upon imbibition."

Comment. Have the authors determined that the monoallelic expression occurs during the 36 hr imbibition period? Has an expression time course been carried out? Is it possible that expression occurs late in seed development before seed dessication occurs?

Summary. The authors have discovered a new cluster of novel imprinted genes that provide significant insights into the mechanisms that regulate an important biological process – the maternal control of seed dormancy. Their data justify the conclusions made in the manuscript. It is likely that further investigation of their parent-of-origin expression will yield novel mechanisms that regulate plant gene imprinting. I recommend that this paper be published in *eLife* after the authors respond to comments and thoroughly rewrite the paper to improve its clarity.

Reviewer #4:

Firstly I think this is a very interesting study and important because it suggests an actual role for imprinting, which is very welcome, and it shows for the first time that imprinting persists into maturity, and that imprinting is dynamic with respect to dormancy state and environmental temperature during seed maturation. The authors clearly rule out any contamination by maternal tissues of the testa which in any case is well known to be dead at maturity.

My main concern with the current narrative is that imprinted genes contain important germination regulators, notably KAI2, but frustratingly none of these are investigated in any detail such that it is shown that imprinting of these genes affects dormancy in any way. So although the manuscript makes claims about a role of imprinting in dormancy control, these remain hypothetical in my view although it is an interesting possibility. There is therefore no clear relationship between the data gathered that shows that dormancy control is maternal (Figure 1), and the action of any imprinted genes, and this creates a somewhat confusing narrative. Thus although it was shown that dormancy state affects imprinting, it is not clear that imprinting affects dormancy, and the author has not satisfactorily ruled out that the maternal tissues are affecting dormancy state in CVI x C24 or CVI x Col crosses, which they are well known to do (Debeaujon et al., 2001). In fact Figure 1 could be removed from the paper without detracting from the study. Proving a role for imprinting in dormancy control could be achieved using kai2 mutants in crosses for instance, but would be a substantial amount of further work.

The opposite is true for storage protein breakdown, where an effect of the maternal allele is clearly shown with very nice data. So there is a part of the study which is really well supported and where imprinting is clearly shown to affect the physiology of germinating seeds. This is a major advance in my view, but ideally should be uncoupled from the dormancy story which with further work could be elaborated into a second interesting manuscript.

---

## [Author Response]

[…]

However, there are four key areas in which substantial revisions are required:

1) Verification of the new imprinting phenomenon.

The report of gene imprinting in mature seeds is novel and exciting, but there are several reasons to be cautious about the findings. a) The imprinted genes reported are almost completely different from known imprinted genes. b) There is little overlap between genes imprinted in dormant and non-dormant seeds. c) The imprinting phenomenon is transient. d) Only one replicate for each cross was used. e) Some of the Sanger sequencing chromatograms used for validation are inconsistent. For instance, in Figure 2, ABCG30 in non-dormant seeds seems mono-allelic and VIM5 in non-dormant seeds seems bi-allelic, unlike stated.

The term “mono-allelic”, which we used only a few times in the first version of the manuscript, no longer appears. We much prefer the term “preferential maternal/paternal allele expression”, which is now used throughout the text. Indeed, strict monoallelic expression is actually a rare phenomenon (e.g. discussed in Mary Gehring, 2013, Annual Rev. Genet (47:187-208). We define preferential maternal, respectively paternal, allele expression when more than 80%, respectively 60%, of the reads come from the maternal, respectively paternal, alleles.

For ABCG30 the two traces overlap, giving the impression that it is monoallelic. For VIM5, you are probably referring to the “ColxCvi dormant” and “CvixCol non dormant” cases. The yellow trace in the “ColxCvi dormant” case is the maternal allele and its height is consistent with the notion that it represents a minor contribution. Similarly the blue trace in the “CvixCol non dormant” is the maternal allele and also a minor contribution. The VIM5 gene, as well as the ABCG30 (together with other genes) have been reamplified from independent endosperm samples and analyzed by MiSeq ([Supplementary-material SD7-data]). They confirm ABCG30’s dormancy-specific preferential maternal allele expression and VIM5’s preferential paternal allele expression.

This kind of inconsistency is also observed in several cases in Figure 2—figure supplement 2.

Please note that by definition only genes maintaining preferential maternal (or paternal) allele expression in both reciprocal crosses are considered to be imprinted. If a gene loses preferential parental allele expression in one of the F1 seeds then we do not consider that it is imprinting (e.g. in the case of SOT12 (At2g03760) we did not observe preferential maternal expression in non-dormant CvixCol F1 seeds).

All the genes shown in Figure 2—figure supplement 2 are now analyzed by MiSeq ([Supplementary-material SD7-data]).

*f) The p-value used to call imprinted genes (0.05) is quite permissive, and likely led to inclusion of false positives.*

We have confirmed imprinting in independent experiments using MiSeq.

g) The RNA-seq results are not always consistent between reciprocal crosses. For example, CP1 has 5683 maternal vs. 17 paternal reads in Col x Cvi, but only 56 maternal vs. 10 paternal reads in Cvi x Col.

We observed that many of the imprinted genes found in dormant seeds happen to have their expression negatively correlating with seed dormancy levels (e.g. see Figure 2 and, for CP1, Figure 3). The differences that you point out are further consistent with this notion: CvixCol F1 seeds, which tend to display dormancy levels more akin to the maternal ecotype Cvi, are more dormant than ColxCvi F1 seeds and express lower maternal allele CP1 expression levels.

h) Only MEGs specific for the dormant dataset were tested by Sanger sequencing.

We have now included the non-dormant class in the revised manuscript.

For these reasons, it is imperative that the imprinting results are thoroughly verified using a robust, quantitative technique. We suggest RT-PCR of multiple genes from both the dormant and non-dormant datasets, using RNA samples different from those used for RNA-seq, followed by sequencing on an Illumina MiSeq instrument (many PCR products can be multiplexed in one lane).

This is now done in the revised version of the manuscript (Figure 2—figure supplement 3, and [Supplementary-material SD5-data]). A total of 19 genes are further studied: 10 dormant-specific MEGs, 5 non-dormant specific MEGs, 2 MEGs found in dormant and non-dormant datasets and 2 PEGs.

Furthermore, we think it is important to verify parent-of-origin specific expression in mature endosperm using a reporter construct. Because creating a new reporter construct would take a long time, we suggest using the published FWA:GFP reporter.

We collaborated with Drs Daichi Susai and Tetsu Kinoshita to perform this experiment. Their results are shown in Figure 2—figure supplement 4. Note that the observed signal is rather weak, likely due to the low *FWA* expression in the Col ecotype background.

A *pFWA::dFWA-GFP* reporter line was reciprocally crossed with a *pDD65::mtKaede* line (labeling mitochondria). Although broad background fluorescence was visible in the cytosol in both directions of cross, some cells showed nuclear localized green fluorescence that was only detectable in the female *pFWA::dFWA-GFP* cross. For some unknown reason Drs D. Susai and T. Kinoshita observed high background noise in Col-0 x *pFWA::dFWA-GFP* and *pFWA::dFWA-GFP* x Col-0 F1 seeds, which prevented them to detect FWA-GFP signals.

To strengthen these data, we have performed RT-PCR to specifically amplify transgenic *FWA:GFP* cDNA in both *pFWA::dFWA-GFP* x *pDD65::mtKaede* and *pFWA::dFWA-GFP* x Col reciprocal crosses. The results clearly show occurrence of preferential maternal allele expression (Figure 2—figure supplement 4).

In addition to new experiments, please provide more information about the analysis of imprinted genes. Specifically, please include in a supplementary table parent-of-origin RNA-seq counts for all genes, not just those deemed significantly imprinted,

The Materials and methods section gives more details and the supplementary table is now provided in [Supplementary-material SD6-data].

and please state in the text whether most genes conform to the expected 2:1 maternal/paternal ratio.

The majority of genes conform to the expected 2:1 maternal/paternal ratio. This is shown in Figure 2—figure supplement 2. Also, we noted that hundreds of genes had a high expression bias when present as either Cvi or Col alleles. This is consistent with previous reports analyzing endosperm and embryo gene expression during early embryogenesis (Nodine and Bartel, 2012, Pignata et al. 2014).

Also, please explain in more detail in the methods section how imprinted genes were called, particularly how information from the reciprocal crosses was used.

The Materials and methods give more details about how imprinted genes were called.

2) A direct functional link between imprinted expression and dormancy.

The reviewers are in agreement that the paper presents evidence for a link between imprinting and dormancy, but not direct evidence that imprinted genes regulate dormancy. The paper shows that maternal control of dormancy cannot be explained by the predominance of maternal genetic material in the endosperm, but this is not proof that imprinted expression controls dormancy. Signals from the maternal seed coat could influence the state of the endosperm, so that even after the seed coat degenerates, how endosperm regulates dormancy may be governed by earlier seed coat signals. Correlation of imprinted gene expression with dormancy can run in either direction: gene expression may influence dormancy, or dormancy can influence gene expression. The gene that was examined in detail, CP1, fits the latter pattern. The effects of the aln mutation were not tested in F1 seeds derived from reciprocal crosses with wild-type, so it remains unclear whether maternal expression of ALN regulates dormancy. Therefore, the title of the paper presently does not accurately reflect the reported findings.

We feel the best way to remedy this would be to show that maternal expression of ALN (or another gene, such as KAI2) regulates dormancy. Depending on the results of these experiments, the paper's conclusions may need to be adjusted to reflect a link with dormancy instead of a causative relationship.

We have performed the suggested experiment by examining the dormancy of *aln* x Col and Col x *aln* F1 seeds, which show maternal inheritance of seed dormancy. Furthermore, similarly as we did for *CP1*, we show that this maternal effect involves gametophytic alleles. The results are shown in Figure 5.

3) Quantification of cruciferin (CRU) proteins.

There is concern that the quantification of CRU proteins reported in this paper may not be reliable. First, the loading control for the CRU assays may be inadequate. In Figure 4 another protein band is used as a loading control, but when looking at Figure 4, this band varies greatly between samples. This raises the question of whether there is indeed a depletion of the putative CRUs during dormancy break, or if the low band intensity is due to less total protein being loaded on the gel. Second, reliance on coomassie-stained gels, instead of Western blots, to determine CRU levels may lead to inaccurate CRU quantification. To address these concerns, please include an adequate loading control, and either utilize Western blots or explain why coomassie staining is a reliable method.

We have repeated all the experiments using new and independent seed material and a specific CRU antibody in Western blots (Figure 4—figure supplement 1 and new Figure 4). For the loading control with use a non-specific band whose intensity remains constant under all developmental and temporal conditions. The new results confirm our previous observations. We have kept the coomassie gel data as a supplementary figure (Figure 4—figure supplement 2).

4) Manuscript readability.

During initial editorial assessment of the manuscript and peer review, a common concern was that the writing presents difficulties even for experts, with descriptions like "circular", "turgid" and "repetitive". Please do your best to improve the logical flow and accessibility of the paper, keeping in mind the broad readership of eLife. To assist you, we are enclosing the original reviews below, which include many suggestions for manuscript improvement.

We are sorry about the bad readability. We have done our best to make it more readable. The Introduction has been rewritten and the Results sections simplified. Of course, we have taken into account the reviewers’ suggestions. We hope you will find that the manuscript is now more readable.

Reviewer #1:

[…]This said, this manuscript has substantial weaknesses. The greatest is the absence of unambiguous evidence that imprinting regulates dormancy. The authors show that maternal control of dormancy cannot be explained by the predominance of maternal genetic material in the endosperm, but this is not proof that imprinted expression controls dormancy. Signals from the maternal seed coat could influence the state of the endosperm, so that even after the seed coat degenerates, how endosperm regulates dormancy may be governed by earlier seed coat signals. Correlation of imprinted gene expression with dormancy can run in either direction: gene expression may influence dormancy, or dormancy can influence gene expression. The gene that the authors chose to examine in detail, CP1, fits the latter patterns. The authors state that several of the imprinted genes are known regulators of dormancy – why didn't they test whether maternal expression in the endosperm of one or more of these is important for dormancy? I think this would be the single most important addition to the paper, which would clearly establish that imprinted expression of a particular gene or genes regulates dormancy.

If by dormancy you only mean “lack of germination” then yes, the first version of our manuscript does not provide unambiguous evidence that imprinting regulates dormancy. However, “germination” is only one of many developmental aspects that characterize dormant seeds. One of the aspects is how CRU protein levels are downregulated according to dormancy levels. The *cp1* mutants can be seen as more “dormant” in what concerns their CRU abundance upon imbibition. The reason we focused on *CP1* is because its (imprinted) expression is remarkably specific to the mature endosperm in imbibed dormant seeds. We reasoned that this would give us the best opportunity to detect parent-of-origin effects and indeed gamete-of-origin effects in CRU protein abundance.

In any case we completely agree with you that it would be preferable to also show that imprinted gene expression of a regulator of gene dormancy can also have consequences for germination. We are now including similar experiments for the case of *ALN*, which further support the view that imprinting regulates dormancy. This time the readout is germination percentage rather than CRU abundance (Figure 5).

The other major weakness, related to the above, is that the manuscript makes for a difficult read. The text is frequently dense, circuitous and repetitive, and the logical thread from dormancy to CP1 is hard to follow.

We hope you will find the new version more readable (see also our comments to the editor and the other reviewers who made a similar critique).

A smaller issue is the lack of any mechanistic insight into how the newly identified genes are imprinted. This is probably beyond the scope of this manuscript, but the authors could at least check if the new imprinted genes are enriched in DME-catalyzed DMRs or PcG targets (as measured by H3K27me3).

We are fully aware that the work leaves aside the mechanistic aspects of the observed imprinting. We are working on this issue and we plan to report about our findings in a future manuscript. We think that mechanistic aspects are beyond the scope of this work.

Reviewer #2:

In this work, Piskurewicz et al. report that maternal regulation of seed dormancy in Arabidopsis may be caused by the differential expression of maternal and paternal alleles in the endosperm (imprinted genes). This finding is novel and exciting; however, I have several concerns that require to be addressed to fully support this major claim.

1) One major concern with this work is the lack of details for the analysis of imprinted genes. Apparently no replicates have been generated, limiting the value of the analysis. It is also not clear how many genes follow the expected ratio of two maternal to one paternal and how many deviate. Furthermore, how many genes are commonly imprinted in reciprocal crosses and how many are not? The data provided in [Supplementary-material SD6-data] do not allow to extract this information, partly because the labelling of the columns is incomplete, partly because only selected genes are shown.

We would rather not discuss the genes that appear to have preferential parental allele expression in only one of the two reciprocal crosses. Indeed, in these cases we do know whether these genes are imprinted or whether they correspond to a gene with an accession expression bias. Nevertheless, these genes are now accessible in the new [Supplementary-material SD6-data].

Other issues are addressed in the new version of the manuscript:

a) RT-PCR products of candidate MEGs and PEGs (dormant and non-dormant case) were performed using independent endosperm material

b) We include a new figure showing that most genes conform to the expected 2:1 maternal/paternal ratio.

c) The labeling of the column was completed

d) The material and method section was improved.

2) To provide convincing proof that genes are specifically maternally expressed in the endosperm after imbibition, reporter genes should be tested. There are published reporters for e.g. FWA that could be used for this analysis.

This experiment is now included in Figure 2—figure supplement 4.

3) The authors do not provide genetic evidence that the identified imprinted genes have a role in seed dormancy. Even if there is indeed a cluster of MEGs whose expression correlates with the dormancy status of different Arabidopsis accessions (and their hybrids), it is not possible to distinguish what is cause and consequence. It is possible that those genes are expressed because dormancy was broken rather than their expression being the cause for the break in dormancy. For instance, the authors propose that the differential expression of CP1, a MEG, may be involved in dormancy due to its potential role in protein storage decay. However, cp1 mutant seeds do not seem to have different dormancy levels compared to WT. Therefore, while there may be a correlation between the expression of certain MEGs and dormancy levels, there is no evidence in this work for a functional link between the two.

Please also see our comments to reviewer #1. *cp1* mutants are “more dormant” from the perspective of CRU protein abundance since they retain higher CRU abundance over time upon seed imbibition relative to WT. If by dormancy you only mean “lack of germination” then we agree with you. However, we tend to view absence of germination as only a particular aspect of dormancy (as stated in the Introduction).

In any case, we have now included a study, similar to the one performed for *CP1*, describing the case of *ALN*, which negatively regulates dormancy from the aspect of germination.

4) I also have major concerns regarding the analysis of cruciferin (CRU) levels in this manuscript. Unlike stated by the authors, Barthole et al. (2014) do not rely on coomassie-stained gels to determine CRU levels, but instead make use of anti-CRU antibodies in a Western-blot. This raises the question of how can the authors in this manuscript distinguish CRUs from other abundant proteins? Furthermore, there is no adequate loading control for the assays. In Figure 4 the authors make use of another protein band for loading control, but when looking at Figure 4, this band varies greatly between samples. This raises the question of whether there is indeed a depletion of the putative CRUs during dormancy break, or if the low band intensity is due to less total protein being loaded on the gel. In my view, these two points raise major questions on whether the CRU-related data is reliable.

We have repeated these experiments using new seed material and a specific antibody. See also our comments to the editor. Results are shown in Figure 4.

5) The chromatograms for the Sanger sequencing are sometimes dubious. For instance, in Figure 2, ABCG30 in non-dormant seeds seems mono-allelic and VIM5 in non-dormant seeds seems bi-allelic, unlike stated. This kind of inconsistency is also observed in several cases in Figure 2—figure supplement 2.

Please see our comments to the editor.

6) Why were only MEGs specific for the dormant dataset tested by Sanger sequencing?

This is due to our particular interest in seed dormancy. We now include more data for the non-dormant dataset (Figure 2, Figure 2—figure supplement 3, and [Supplementary-material SD5-data]).

7) The effect of aln on dormancy should be tested in F1 seeds derived from reciprocal crosses with wild-type to test whether the mutant phenotype has a maternal origin.

We performed these experiments and we do indeed detect maternal effects. Please see Figure 5for details.

Reviewer #3:

Recently harvested seeds are often dormant and fail to germinate when they are imbibed. It is well-known that the maternal parent plays a major role in establishing dormancy, and that mature endosperm, a single layer of cells between the embryo and maternal testa cell layers, is essential for dormancy. Different Arabidopsis accessions display different levels of seed dormancy. C24 has low dormancy and Cvi has high dormancy, and the differences are quite dramatic. After a short 5-day after-ripening period, neither C24 or Cvi seeds germinate after 72 hrs imbibition. After an intermediate after-ripening period (25 days), full germination of C24 occurs, while almost no Cvi seeds germinate. Both C24 seeds and Cvi seeds after a long after-ripening period (6 months). The authors used these dormancy differences between C24 and Cvi to show that:

The maternal parent determines the level of seed dormancy. That is, F1 seed from C24xCvi have low seed dormancy to C24, and F1 seed from CvixC24 have high seed dormancy equivalent to Cvi.

The maternal bias is not due to the endosperm's 2:1 ratio of maternal:paternal chromosomes. That is F1 progeny C24xCvitetraploid and CvixC24tetraploid had similar levels of dormancy as F1 progeny between diploid parents.

These results motivated them to determine whether gene imprinting might underlie the maternal inheritance of seed dormancy.

Comment. Throughout the manuscript, the authors use the word "gene imprinting". However, the issue of whether a gene imprint exists that distinguish maternal versus paternal alleles is never addressed. Therefore, although it is OK to use the phrase gene imprinting, I think the phrase parent-of-origin gene expression should also be incorporated into the paper.

OK, we introduced this phrase in the Introduction as follows:

“Imprinting gene expression, also called genomic imprinting, is the preferential expression of a given parental allele over the other. Such parent-of-origin gene expression is observed in both mammals and flowering plants, which share the habit of nourishing the embryo through a sexually derived tissue (Pires and Groniklaus, 2014).”

The authors convincingly show that the seed coat (endosperm + testa) can be used to isolate endosperm RNA that will be used to identify parent-of-origin gene expression in the endosperm because the testa contributes little mRNA.

For the high-throughput RNA-seq experiments, the authors replaced C24 with Col, which is considered to be a low dormancy accession. Following a short after-ripening period of 10 days, 40% of ColxCvi F1 seeds germinated whereas only 1% of CvixCol F1 seeds germinated. (Figure 1). When intermediately after-ripened for 20 days, about 80% of ColxCvi F1 seeds germinated whereas only 10% of CvixCol F1 seeds germinated.

Comment. In the Material and Methods section, the authors state, "ColxCvi and CvixCol seeds were harvested the same day and after-ripened for 10 days (dormant seeds) or for 2 months (non-dormant seeds). Total RNA samples were isolated from seed coats (n=200) dissected 36 hours after seed imbibition."

This sentence should be moved into the body of the text because it clearly defines what is meant by "dormant seeds" and "non-dormant" seeds, terminology that is first used in subsection “Dormancy-specific genomic imprinting in the endosperm” paragraph seven for "dormant" CvixCol and ColxCvi F1 seeds, and "non dormant" CvixCol and ColxCvi F1 seeds.

In the first submitted version of the manuscript we referred to the dormancy of CvixCol and ColxCvi F1 seeds slightly earlier (“RNA was extracted from seed coats dissected from dormant and non-dormant CvixCol and ColxCvi F1 seeds 36h upon seed imbibition (Figure 2).”).

However, we agree with you that it is helpful to better define in the main text how we generated this seed material. In the second submitted version of the text we write: “*ColxCvi and CvixCol seeds were harvested the same day and after-ripened for 10 days (dormant seeds) or for 2 months (non-dormant seeds). Total RNA samples were extracted from the endosperm (n=200) of dormant and non-dormant CvixCol and ColxCvi F1 seeds 36h upon seed imbibition (Figure 2).*”

It should be noted that after 10 days of ripening, 40% of the ColxCvi F1 seeds germinated and 60% did not, so they are a mixture of approximately equal numbers of dormant and non-dormant seeds. Perhaps the authors could explain why this time point was chosen.

Please note that in Figure 1, germination is scored at 72h. In reality all seeds are still dormant as they did not germinate at earlier time points (36 hours). Overall this seed population is still more dormant than Col.

Furthermore, we knew from previous experiments that certain genes regulating dormancy are not detectable in freshly harvested seeds (no after-ripening) relative to more after-ripened seeds (e.g. as in the case of *CP1*). We therefore reasoned that by choosing a substantial amount of after-ripening time that is partially breaking dormancy we would have a higher chance to identify imprinted dormancy genes.

Although it is likely that at 2 months all ColxCvi and CvixCol F1 seeds are non-dormant, the authors never show data supporting this, and only present the 6-month after-ripening data (see above).

The reason we show 6 months of after-ripening in Figure 1 is because that is what is needed in order to break Cvi dormancy. Indeed, the ColxCvi and CvixCol F1 seeds are no longer dormant after 2 months.

The authors identified candidate imprinted genes by high-throughput sequencing RNA from F1 hybrid seeds; using SNPs to distinguish parent-of-origin expression.

They identified 67 MEGs/4 PEGs in dormant seed coat (endosperm) and 49 MEGs/8 PEGs for non-dormant seed coat (endosperm). Only 14 MEGs were present in both dormant and non-dormant seed coat. Only 5 MEGs and 3 PEGs have been previously identified in studies using seeds at earlier stages of development.

9 of the MEGs were selected for further analysis. Sanger sequencing experiments using F1 seed from reciprocally crossed parents was used to verify parent-of-origin expression. Interestingly, the parent-of-origin specific expression was dynamic – with many showing parent-of-origin expression at the dormant stage (10-day after-ripening) and biallelic expression at the non-dormant stage (2-month after-ripening).

As shown in Table 3, many of the dormancy-specific MEGs likely play roles in regulating dormancy/germination. The authors focused on the function of two particularly interesting MEGS – ALN and CP1.

ALN. As described in Watanabe et al. (2014, Plant Cell Environment, 37:1022) the purine metabolite, allantoin, promotes abscisic acid (ABA) production by activating transcription of a key enzyme for ABA biosynthesis, and by a post-translational activation mechanism. ABA is a major plant hormone that promotes dormancy. The ALN gene encodes allantoin amidohydrolase, which, according to Werner et al. (Plant Physiol 2008, 146:418) degrades allantoin. Consistent with this, aln loss-of-function mutations result in increased allantoin production, and increased ABA production.

Comment. I believe that the function of the ALN gene can be more clearly described in the manuscript. Also, only the Watanabe reference is cited, but it is not included in the references.

We have now included more information about Allantoinase by describing its house-keeping role in the purine degradation pathway as well as its emerging role in the plant’s responses to stress. In this context ALN was shown to regulate ABA synthesis by Watanabe et al. 2014. This reference is now included in the list of references. Thank you.

Importantly, the authors show that the aln mutant displays high dormancy relative to wild-type seeds. Thus, the ALN MEG gene is a positive regulator of dormancy.

The authors next raise the issue whether there is an ecotype bias for expression of the MEGs.

Comment. Analysis of expression after reciprocal crosses seemed to point to expression of alleles being controlled by their parent-of-origin, not by their respective ecotypes. So, it seemed that the issue of ecotype bias was already ruled out for gene imprinting.

However, the authors investigated ecotype-biased expression by assessing whether dormancy-specific MEG expression correlates with dormancy levels imposed by a given ecotype. They show that somewhat less than half (23 out of 53) had expression levels that negatively correlate with seed dormancy levels. That is, lower expression is correlated with higher dormancy.

Comment. I believe that the authors are not addressing an ecotype effect that affects the silencing of the paternal allele (gene imprinting). Rather, their clustering results point to a group of genes that conform to the ALN gene model, where lower maternal expression results in higher dormancy (see above).

We are not sure we understand your comment. We think there are two things that need to be distinguished:

Firstly, imprinted genes are those whose preferential parental allele expression is observed in both sets of seeds arising from reciprocal crosses. Thus, the identification of an imprinted gene is independent of its expression levels (of course genes with too low expression are discarded for this analysis).

Secondly, one can next be interested by the expression of these genes according to environmental cues (e.g. cold) or ecotype. Our point is that a substantial fraction of dormancy-specific MEGs happens to have high expression in Col, which has low dormancy levels, and low expression in Cvi, which has high dormancy levels. Furthermore, cold, which increases dormancy in both ecotypes, diminishes expression in both cases. These observations suggested to us (our model) that the maternal allele regulates dormancy levels by imposing expression levels of dormancy genes that is characteristic of the maternal ecotype. These levels would be further regulated by temperature during seed development.

CP1.

*The CP1 gene encodes cysteine protease1, which promotes storage protein decay. CP1 expression is activated upon imbibition, resulting in release of amino acids from storage proteins to be used by the young seedling. CP1 is a MEG whose expression level increases with after-ripening time, but the level of CP1 expression is lower in Cvi (high dormancy) than in Col (high dormancy). Thus, CP1 expression, like ANL and the gene cluster (see above) is negatively correlated with dormancy.*

The authors go on to show that the CP1 allele inherited from the maternal parent, and more specifically, the CP1 allele inherited from the maternal gametophyte, promotes decay of Arabidopsis seed storage proteins, cruciferins (CRU).

Finally, expression of CP1, like ANL, becomes biallelic as ripening time passes. Thus, the parent-of-origin expression is dynamic.

Discussion.

The authors state, "Concerning the MEGs and PEGs identified in non-dormant seeds it seems likely that their monoallelic transcription takes place upon seed imbibition, since they were not identified in dormant seeds. It appears therefore that a dedicated developmental program of genomic imprinting is operating in mature seeds upon imbibition."

Comment. Have the authors determined that the monoallelic expression occurs during the 36 hr imbibition period? Has an expression time course been carried out? Is it possible that expression occurs late in seed development before seed dessication occurs?

In the first version of the manuscript (Figure 4—figure supplement 6, now Figure 4—figure supplement 8) we monitor *CP1* mRNA abundance over time upon seed imbibition (time points: 24h, 48h, 72h). The expression increases while remaining preferentially maternal. This strongly suggests that at least for the case of *CP1*, imprinted gene expression occurs de novo upon imbibition and therefore is not only a carry over from developing seeds.

We did not monitor *CP1* expression during seed development and we cannot rule out that *CP1* is expressed during seed development. However, when seeds are dormant, *CP1* expression is undetectable. This rules out that the detected *CP1* expression corresponds to that already present in dry dormant seeds.

Summary. The authors have discovered a new cluster of novel imprinted genes that provide significant insights into the mechanisms that regulate an important biological process – the maternal control of seed dormancy. Their data justify the conclusions made in the manuscript. It is likely that further investigation of their parent-of-origin expression will yield novel mechanisms that regulate plant gene imprinting. I recommend that this paper be published in eLife after the authors respond to comments and thoroughly rewrite the paper to improve its clarity.

We hope the new version is now more readable.

Reviewer #4:

[…]

My main concern with the current narrative is that imprinted genes contain important germination regulators, notably KAI2, but frustratingly none of these are investigated in any detail such that it is shown that imprinting of these genes affects dormancy in any way. So although the manuscript makes claims about a role of imprinting in dormancy control, these remain hypothetical in my view although it is an interesting possibility. There is therefore no clear relationship between the data gathered that shows that dormancy control is maternal (Figure 1), and the action of any imprinted genes, and this creates a somewhat confusing narrative. Thus although it was shown that dormancy state affects imprinting, it is not clear that imprinting affects dormancy, and the author has not satisfactorily ruled out that the maternal tissues are affecting dormancy state in CVI x C24 or CVI x Col crosses, which they are well known to do (Debeaujon et al., 2001). In fact Figure 1 could be removed from the paper without detracting from the study. Proving a role for imprinting in dormancy control could be achieved using kai2 mutants in crosses for instance, but would be a substantial amount of further work.

We would like to keep Figure 1 since it is the observation that initiated the study. We use this figure to explain the different scenarios that could explain our observations, which include gene dose effects, maternal tissue effects and imprinting.

A role for imprinting in dormancy was strongly suggested by our experiments with *aln* mutants (see Figure 5).

The opposite is true for storage protein breakdown, where an effect of the maternal allele is clearly shown with very nice data. So there is a part of the study which is really well supported and where imprinting is clearly shown to affect the physiology of germinating seeds. This is a major advance in my view, but ideally should be uncoupled from the dormancy story which with further work could be elaborated into a second interesting manuscript.

See our comments to the other reviewers above).